# Genome evolution and the emergence of pathogenicity in avian *Escherichia coli*

Leonardos Mageiros [1], Guillaume Méric [1], Sion C. Bayliss[1,2], Johan Pensar[3,4], Ben Pascoe [1,3], Evangelos Mourkas [1], Jessica K. Calland[1], Koji Yahara [5], Susan Murray[6], Thomas S. Wilkinson [7], Lisa K. Williams[7], Matthew D. Hitchings[7], Jonathan Porter[8], Kirsty Kemmett[9], Edward J. Feil[1], Keith A. Jolley [10], Nicola J. Williams[9], Jukka Corander [3,4,11] & Samuel K. Sheppard [1,2,10 ✉]

Chickens are the most common birds on Earth and colibacillosis is among the most common diseases affecting them. This major threat to animal welfare and safe sustainable food production is difficult to combat because the etiological agent, avian pathogenic *Escherichia coli* (APEC), emerges from ubiquitous commensal gut bacteria, with no single virulence gene present in all disease-causing isolates. Here, we address the underlying evolutionary mechanisms of extraintestinal spread and systemic infection in poultry. Combining population scale comparative genomics and pangenome-wide association studies, we compare *E. coli* from commensal carriage and systemic infections. We identify phylogroup-specific and species-wide genetic elements that are enriched in APEC, including pathogenicity-associated variation in 143 genes that have diverse functions, including genes involved in metabolism, lipopolysaccharide synthesis, heat shock response, antimicrobial resistance and toxicity. We find that horizontal gene transfer spreads pathogenicity elements, allowing divergent clones to cause infection. Finally, a Random Forest model prediction of disease status (carriage vs. disease) identifies pathogenic strains in the emergent ST-117 poultry-associated lineage with 73% accuracy, demonstrating the potential for early identification of emergent APEC in healthy flocks.

---

[1] The Milner Centre for Evolution, Department of Biology and Biochemistry, University of Bath, Claverton Down, Bath, UK. [2] MRC Cloud Infrastructure for Microbial Bioinformatics (CLIMB) Consortium, London, UK. [3] Department of Biostatistics, University of Oslo, Oslo, Norway. [4] Department of Mathematics and Statistics, Helsinki Institute for Information Technology, University of Helsinki, Helsinki, Finland. [5] Antimicrobial Resistance Research Centre, National Institute of Infectious Diseases, Tokyo, Japan. [6] Uppsala University, Department for medical biochemistry and microbiology, Uppsala University, Uppsala, Sweden. [7] Swansea University Medical School, Institute of Life Science, Swansea SA2 8PP, UK. [8] National Laboratory Service, Environment Agency, Starcross, UK. [9] Department of Epidemiology and Population Health, Institute of Infection & Global Health, University of Liverpool, Leahurst Campus, Wirral, UK. [10] Department of Zoology, University of Oxford, South Parks Road, Oxford OX1 3PS, UK. [11] Parasites and Microbes, Wellcome Sanger Institute, Cambridge, UK. ✉email: s.k.sheppard@bath.ac.uk

A seemingly insatiable human appetite for poultry meat and eggs has resulted in modern livestock farming on a colossal scale. Today there are over 26 billion chickens worldwide, with poultry constituting around 70% of all bird biomass on earth[1]. Advances in selective breeding and husbandry have greatly increased productivity in the last 50 years but this level of agricultural intensification brings significant challenges for animal health, welfare and safe sustainable food production. Of particular concern are the opportunities created for the spread of livestock diseases and the emergence of zoonotic pathogens[2,3].

Among the most common bacterial diseases of chickens reared for egg and meat production is colibacillosis[4] caused by avian pathogenic *Escherichia coli* (APEC). Like other forms of extraintestinal pathogenic *E. coli* (ExPEC)[5], APEC exists as a commensal component of the avian gut microbiota but emerges to cause a variety of systemic infections. Diseases of chickens and other birds range from epidermal, yolk sac and common respiratory tract (aerosacculitis) infections, to severe pericarditis, perihepatitis, omphalitis and septicaemia[6,7]. In some cases mortality can reach 20%, condemning whole flocks, leading to suffering for millions of farmed birds and multimillion pound losses to the worldwide poultry industry[4,6]. The problem is exacerbated by the rise of antimicrobial resistance occurring across global transmission networks[8–12], and the recognition that APEC may cause human infections[13–16] highlights the need to control this bacterium for both animal and human health.

Risk factors for colibacillosis have been identified, and include chicken immunological immaturity and stress[17], but the disease has proved difficult to control, not least because no single gene, plasmid, phage or pathogenicity island has been exclusively associated with the emergence of virulent APEC from a background of harmless gut-dwelling *E. coli*[18,19]. Technical advances in high-throughput whole-genome sequencing offer opportunities to investigate the population genomics of pathogen evolution[20] but understanding the spread of APEC remains challenging for two reasons. First, there is uncertainty about the extent to which disease results from the transmission of a few globally distributed epidemic clones[4,6,7] or a diverse assemblage of disease-causing lineages[21,22]. Second, while a considerable body of knowledge has been gathered[10,23–26], the genes contributing to APEC virulence are less well described than in human ExPEC pathotypes[27,28]. Pathogenicity is often linked to the presence of plasmids that confer a range of virulence-associated traits[10,23–26], such as aerobactin production, complement resistance and iron acquisition[7,10,22]. However, no one gene is known to be essential for the development of extraintestinal infection in birds[10,29–31] and pathogenicity appears to be linked to a heterogenous mix of plasmid and chromosomal genes involved in bacterial adhesion, invasion, toxicity, antibiotic resistance, survival and metabolism under stress[22,31–33].

With colibacillosis set to increase in line with expanding poultry production there is a pressing need to monitor the emergence of APEC within genetically diverse commensal populations and identify strains that are predisposed to pathogenicity because of the genetic elements harboured in their genome. Here we take a large-scale comparative genomics approach to investigate the genetic basis of APEC pathogenicity that is agnostic to pre-existing assumptions about putative virulence determinants. Using a genome-wide association study (GWAS) approach[34,35], we analyse 568 *E. coli* genomes from commercial poultry farms, including isolates from healthy chickens and those from various systemic infection body sites, and identify genes and genetic elements associated with avian pathogenicity (Fig. 1). Finally, having described an evolutionary context for understanding pathogen emergence, we use a machine learning approach to identify risk genotypes, that with further validation, could form a basis of diagnostics and interventions to improve animal health.

## Results

**Core and accessory genome variation in avian *E. coli*.** The pangenome of the 568 avian *E. coli* isolate dataset (309 disease-associated and 234 asymptomatic carriage strains) comprised 15,281 unique genes, with an average of 4115 genes per isolate. These included 3094 genes present in at least 95% of the dataset, which corresponded to 75% of the average genome size, consistent with previous *E. coli* core genome estimates[28,36]. The rate of accessory gene discovery did not plateau as the sampling increased (Supplementary Fig. 1), consistent with widespread acquisition of genes through horizontal gene transfer (HGT). While only 15.5% of all annotated genes from the reference avian *E. coli* strain APEC_O1 were of unknown function, this number increased to 65.8% for the whole pangenome. All the assembled genomes analysed in this study are available via Figshare (https://doi.org/10.6084/m9.figshare.12011811) and raw sequence data has been deposited in the sequence read archive (SRA) associated with BioProject PRJNA592536.

**Quantitative analysis of APEC plasmid genes does not fully explain pathogenicity.** The emergence of APEC has been widely linked to the acquisition of plasmids containing virulence genes[10,24–26,32]. Therefore, we first quantified the presence of putative plasmid genes in isolates associated with disease and asymptomatic carriage using a gene-by-gene approach[37,38]. Putative plasmid genes were widely distributed among APEC and commensal *E. coli* strains in chickens and there was evidence that the average number of plasmid genes per isolate was greater among commensal strains (Supplementary Fig. 2). These results provide initial evidence that emergence of APEC virulence is not entirely dependent on the presence of specific defined plasmids, as in some *E. coli* pathotypes[39]. In fact, rather than a pattern of complete plasmid (as detailed in the reference sequences) presence/absence, there was evidence for mosaicism of plasmid genes found together in different combinations. Furthermore, these analyses do not discriminate the context of putative plasmid-associated genes that may be plasmid-borne or integrated in the chromosome. One explanation for the high numbers of putative plasmid genes harboured among commensal isolates is that some contribute to avian adaptation rather than sensu stricto virulence. For example, plasmid-borne antimicrobial resistance genes[40] may promote persistence in intensive livestock systems, where antimicrobials may have been used for prophylaxis or treatment, but are not directly associated with invasive disease. However, while some putative plasmid genes were more common in APEC compared to other *E. coli*, there was little evidence of complete segregation that would be indicative of direct causation (Fig. 2 and Supplementary Fig. 2). These findings suggest that a full understanding of the genetic determinants of APEC emergence requires consideration of homologous sequence variation rather than simple plasmid gene presence/absence analysis.

**Avian pathogenic strains emerge from multiple *E. coli* lineages.** A maximum-likelihood phylogeny constructed from a concatenated gene-by-gene core genome alignment (3,094 genes) revealed a highly structured population (Fig. 3a). Inclusion of isolates from the well-described ECOR collection allowed contextualisation of avian isolates among known *E. coli* phylogroups and multi-locus sequence types (STs)[41,42]. Poultry isolates from our dataset (Supplementary Data 1) clustered within the eight known *E. coli* phylogroups (A, B1, B2, C, D, E, F and G)[43–45]. There was evidence for variation in the distribution of poultry

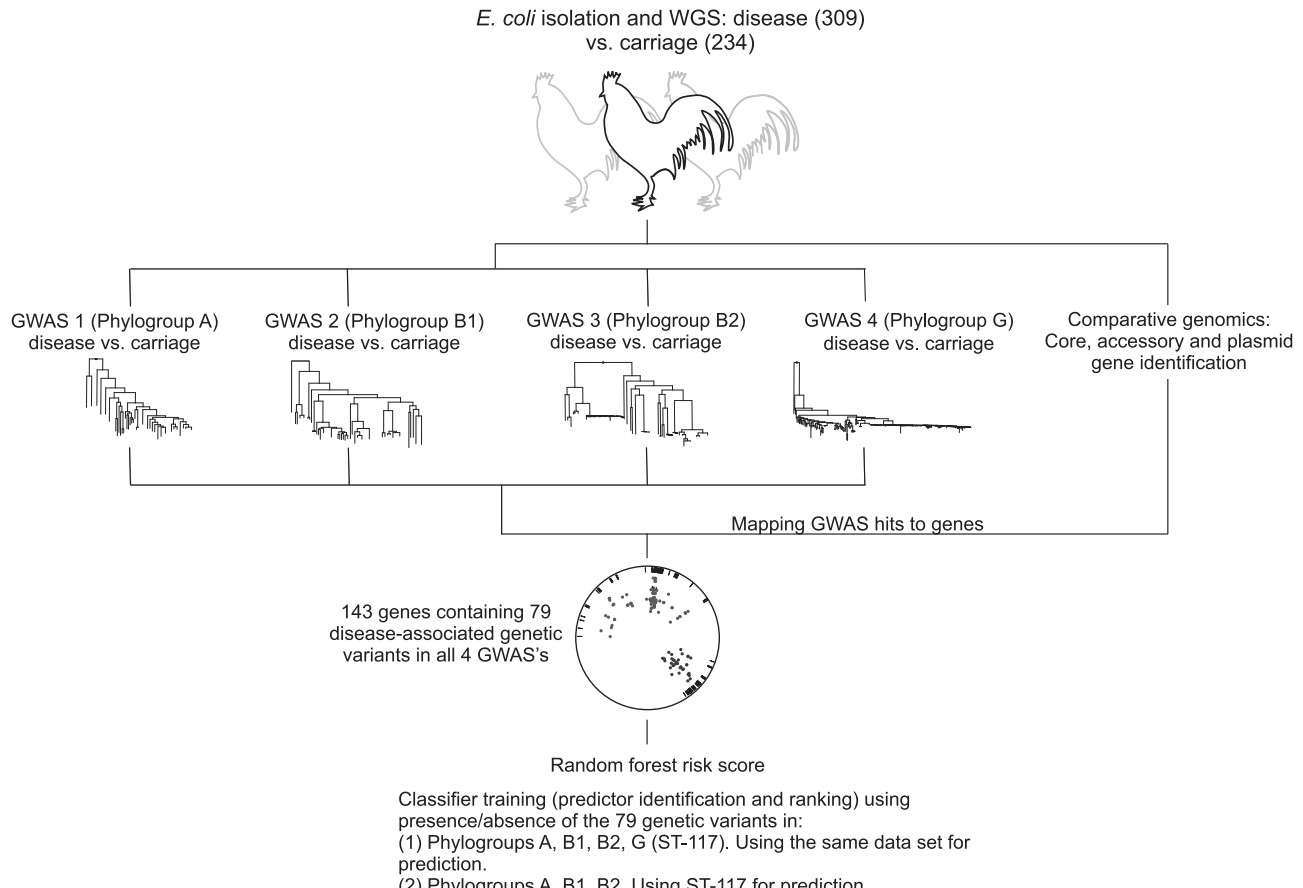

**Fig. 1 Avian Pathogenic _E. coli_ (APEC) GWAS and risk prediction.** Genome-wide association studies (GWAS) can identify multiple genetic variants associated with complex traits but these can be difficult to interpret. For example, pathogenicity is a multifactorial phenotype, potentially involving genes that affect phenotypes like toxicity, antimicrobial resistance, immune evasion etc. Furthermore, the role of certain genes may be poorly defined, especially in bacteria with large accessory genomes. We developed a method in which 4 GWAS experiments (carriage vs. disease isolates) were conducted and the disease-associated genetic variants (core genome SNPs, accessory genes, fission/fusion, duplications and accessory gene alleles) were mapped to genes within the pan genome. Disease-associated elements identified in all four lineage-specific GWAS (phylogroups A, B1, B2 and ST-117 (phylogroup G)) included 143 genes, containing 79 species-wide genetic variants. Patterns of presence and absence of these variants were used as classifiers in two different random forest models to identify the best predictors of APEC disease.

strains among lineages. For example, 211 isolates (39%) belonged to a single sequence type (ST-117) which, together with isolates in the B2, B1 and A phylogroups, constituted 93% of the poultry isolates in our dataset. Of the remaining isolates, the most common STs were: ST-1618, ST-95, ST-919 and ST-429 (phylogroup B2); ST-101, ST-155 and ST-469 (phylogroup B1); ST-38 and ST-69 (phylogroup D); and ST-10 (phylogroup A). Phylogroups E and F were exclusively populated with ST-350 and ST-648 strains respectively. While variation in the frequency of poultry isolates in different lineages may reflect natural abundance within host populations, this does not necessarily indicate pathogenicity. To assess this, the ratio of invasive to commensal strains in the common lineages was determined for each common phylogroup and ranged from 61% (B2) to 31% (E). While it remains possible that lineages with enhanced pathogenicity may exist or emerge in the future, among known _E. coli_ diversity, isolates from all major phylogenetic groups were represented in both the asymptomatic and the disease isolate collections. This reflects the emergence of pathogenic clones from multiple genetic backgrounds.

**Pangenome-wide association study reveals pathogenicity-associated genes.** GWAS was performed for each of the four most common lineages in the dataset, namely on phylogroups A,

B1, B2 and ST-117. In each case, disease isolates were compared to those from asymptomatic carriage within the same phylogroup. The GWAS approach incorporated a ClonalFrameML phylogeny that accounts for the impact of recombination, thereby reducing the effect of population structure and maximising the chance of identifying elements associated with a switch from commensal to pathogenic lifestyle. These independent GWAS analyses identified 11,947, 15,670, 43,980 and 7110 associated genetic elements with a $p$-value < 0.05, that mapped to 1925, 2099, 3946 and 554 infection-associated genes in phylogroup A, B1, B2 and the ST-117 lineage respectively. Nonetheless, only 896 genes contained associated genetic elements with $p$-value < 0.01 (dots on Fig. 3b) and only these were considered for further downstream analysis. Out of the 896 pathogenicity-associated genes, 753 were phylogroup- or lineage-specific, suggesting multiple independent pathways to pathogenicity (Supplementary Fig. 3), with some variation in prevalence based upon extra-intestinal isolation source (Supplementary Fig. 5a/Supplementary Data 7). However, 143 genes were flagged as pathogenicity-associated in all four GWAS analyses (Supplementary Data 2 and Supplementary Fig. 3). Of these, 65 were core genes (45.5%) and 78 accessory genes (54.5%), and had diverse predicted functions, including genes involved in metabolism, lipopolysaccharide

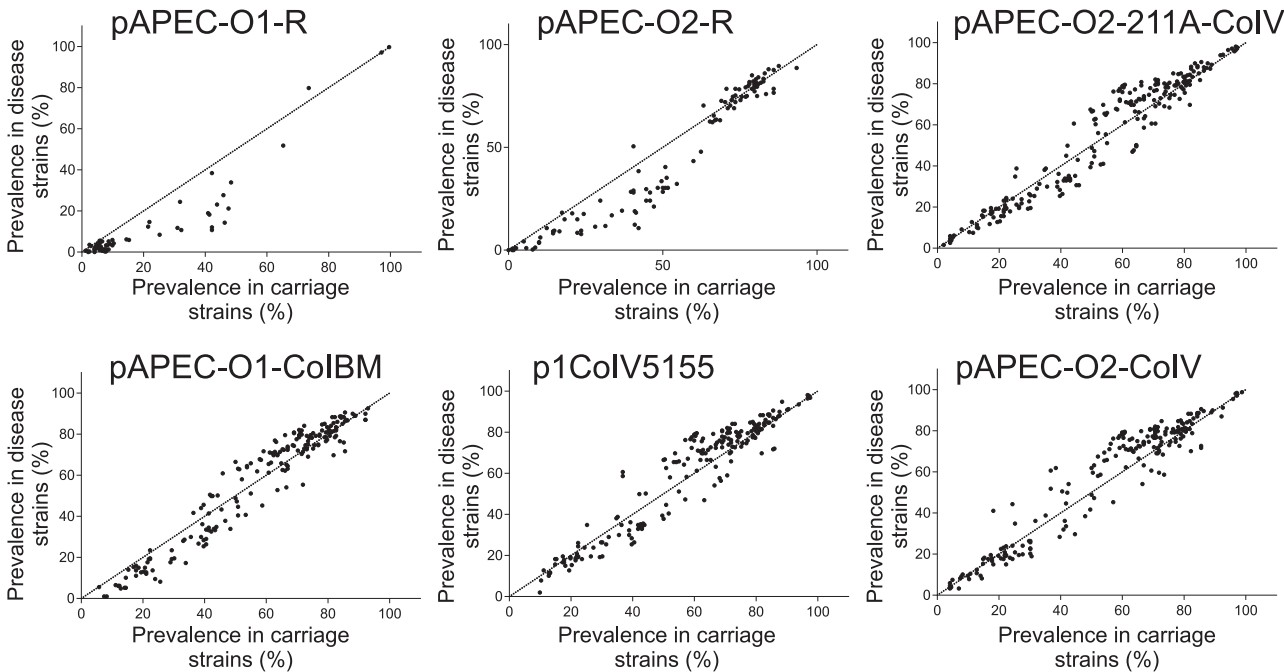

**Fig. 2 Segregation of genes in six known APEC plasmids among carriage and disease isolates.** For every gene (represented with a dot) within six known APEC plasmids, we calculated the prevalence (%) among carriage (*n* = 234) and disease-associated (*n* = 309) *E. coli* genomes. The dashed line represents equal prevalence in each population. Source data are provided as a Source Data file.

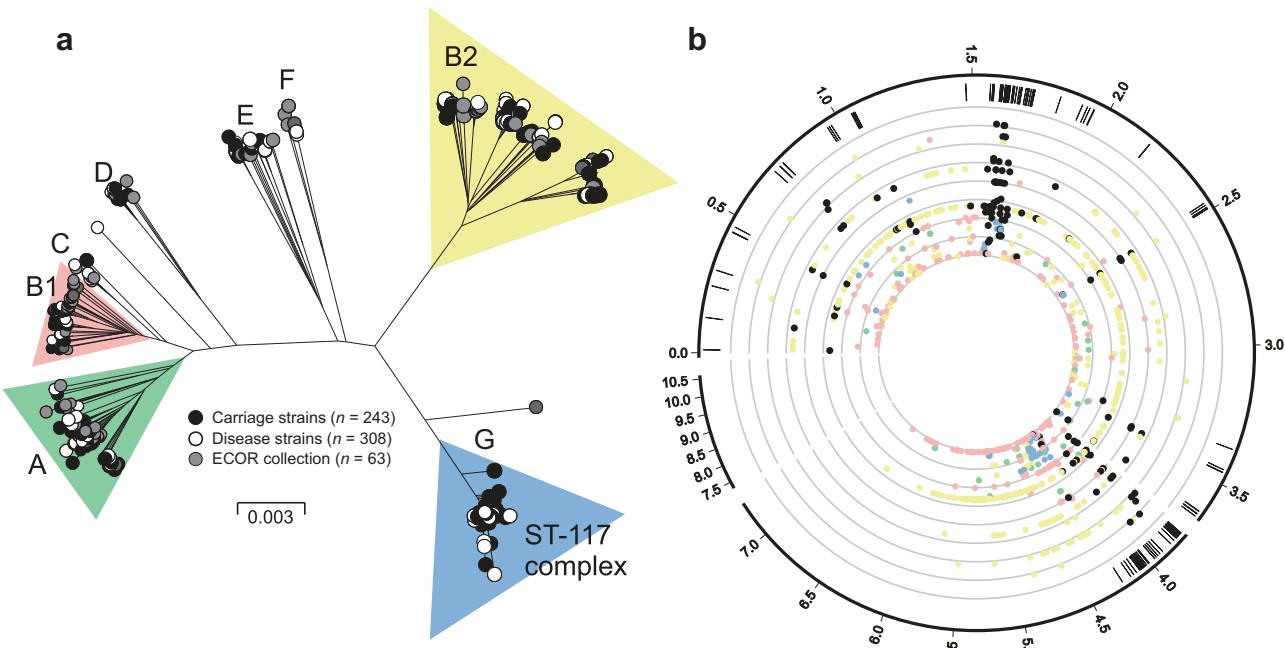

**Fig. 3 Population structure and genome-wide association study of avian *E. coli*. a** Phylogenetic tree of 568 avian *E. coli* strains, reconstructed using a maximum-likelihood algorithm (IQ-TREE) from a core genome alignment (*n* = 3094 genes shared at least by 95% of all the isolates). Isolates are labelled according to source: disease isolates (white); carriage isolates (black); ECOR collection (grey). The letters designate the different *E. coli* phylogroups. **b** Pangenomic position of GWAS results. The outer ring represents the pangenomic position of genes in the *E. coli* reference strain APEC_O1, the rest of the pangenome inferred in this study and a group of low frequency accessory genes that were excluded from the GWAS. Black ticks in the second ring show the position of genes containing disease-associated genetic variants in all four distinct GWAS. Coloured circles are shown for the most statistically associated (lowest *p*-value) elements in a given gene for GWAS in phylogroups A (green), B1 (pink), B2 (yellow), ST-117 complex (phylogroup G; blue) and species-wide disease-associated elements (black). The threshold for significance was *p*-value = 0.01 (inner circle). Statistical significance was determined by the treeWAS algorithm. Concentric rings emanating from this threshold correspond to incremental reductions to a *p*-value of 0.000001 (outer ring). The numbers in the outer ring denote the length of the pangenome in Mbp. Source data are provided as a Source Data file.

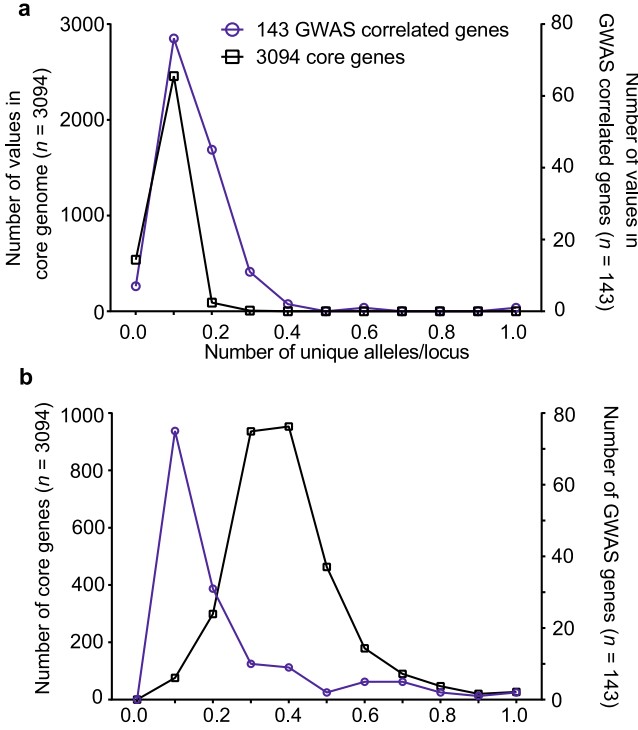

**Fig. 4 Comparison of allelic variation and consistency index for core genes and genes containing disease-associated elements. a** The average number of alleles per locus and **b** consistency indices to a core phylogeny, were calculated for each gene alignment for core genes and 143 genes containing pathogenicity-associated elements using R and the phangorn package. The left *y*-axis indicates the number of core genes (black line), the right *y*-axis indicates the number of genes containing pathogenicity-associated elements (blue line). For the consistency index, the two distributions were significantly different (two-tailed Mann–Whitney test; *p*-value = 0.0001, Mann–Whitney U = 11,366). Source data are provided as a Source Data file.

synthesis, heat shock response, antimicrobial resistance and toxicity. A total of 58 (74.4%) of the accessory genes were putatively of plasmid or phage origin (Supplementary Data 2). Finally, within the 143 species-wide pathogenicity-associated genes, we identified 79 genetic elements that segregated by disease/carriage (*p*-value < 0.05) in all the GWAS, including 66 core genome SNPs, 3 accessory genes, 1 fission/fusion, 4 duplications and 5 accessory gene alleles (Supplementary Data 3). These stringently defined elements constitute robust candidates for disease risk prediction.

**Pathogenicity-associated genes recombine among *E. coli* lineages**. The acquisition of pathogenicity-associated elements among divergent lineages and the importance of potentially mobile elements (plasmid and phage genes) suggest a role for HGT in the emergence of avian pathogenicity. There was a significant increase (Mann–Whitney test; U = 80516, *p*-value < 0.0001) in allelic variation among genes associated with pathogenicity (Fig. 4a), with an average of 0.156 (±0.106) unique alleles per locus for the 143 pathogenicity-associated genes, and 0.08 (±0.035) for 3094 core genes. This could be due to the accumulation of deleterious mutations resulting from *E. coli* range expansion into the pathogenic niche[46] but an equally likely explanation is that these substitutions result from elevated recombination among pathogenicity-associated genes. Evidence for this comes from calculation of the mean consistency index (CI) that was significantly lower

(Mann–Whitney test; U = 11,366, *p*-value < 0.0001) among pathogenicity-associated genes (0.2226 ± 0.2037) compared with other core genes (0.3895 ± 0.01186; Fig. 4b). This suggests that the clonal mode of descent is disrupted in pathogenicity-associated genes consistent with elevated HGT.

**Machine learning identifies disease risk genotypes**. Quantitative determination of species-wide pathogenicity risk markers was carried out using a Random Forest (RF) classifier approach based on the presence/absence of the 79 genomic variants, that were associated with disease isolates in all four lineage-specific GWAS analyses (Supplementary Data 3). Using disease-associated elements found in all major phylogroups maximised the likelihood of capturing generalised predictors of APEC pathogenesis and limited the possible linage specific effects. The estimated risk score was defined as the probability of an isolate coming from disease given a certain profile of the 79 genetic elements. The relative predictive power of each of these elements was estimated by ranking them according to their estimated importance as classifiers in the model (Fig. 5a). Next, the diagnostic ability of the classifier system at varying discrimination thresholds (receiver operating characteristic - ROC curves) and the importance of the 10 highest ranked predictors were investigated in two analyses (Fig. 5b–e). In the first analysis, in which the training data contained all isolates from the four phylogroups, the RF model reached an out-of-sample classification accuracy of 76.9% for predicting infection status of *E. coli* strains (healthy carriage vs. disease). SNPs within the *gnd* gene, involved in inter-strain transfer and recombination[47], accounted for 4 of the 10 most important predictors. These 10 predictors achieved a classification accuracy of 73.5% on their own, potentially offering simple targets for investigation of *E. coli* pathogenicity risk.

In a second analysis, we tested the ability of the model trained on data from phylogroup A, B1 and B2 isolates, to predict infection status (healthy carriage vs. disease) in the emergent ST-117 lineage (phylogroup G) that is thought to be virulent in birds and hold zoonotic potential[48–50]. Replicate analyses with ST-117 isolates included ('Train') and excluded ('Test') from the training data returned seven of the same top ten ranked predictors as in the first analysis (Fig. 5c, e) and gave average RF out-of-sample accuracy of 75% and 73% respectively. The slight reduction in the accuracy when moving outside the training domain of the RF model (area under the ROC 0.79 to 0.76, Fig. 5d), indicated that the model may generalise to other *E. coli* data using existing predictors. Achieving this level of prediction accuracy has clear potential for the development of pathogenicity biomarkers in a farm setting, particularly as the power of the model is limited by the input data. Specifically, as samples from asymptomatic chicken may include strains that have the potential to cause future infection (as well as those that do not), the commensal strain training dataset likely includes some pathogenicity elements. While it is difficult to predict which strains these are, especially as host factors may influence infection, broader sampling may increase the numbers of representative commensal strains that do not have the potential to cause disease, and elevate prediction accuracy beyond 75–77%.

**Prevalence of APEC-associated genetic variants in *E. coli* isolates from different infection sites and other host sources**. The prevalence of APEC-associated variants, used in the RF model, was investigated in isolates sampled from other host niches. Specifically, we analysed the *E. coli* reference collection (ECOR)[42], 175 human ExPEC strains[28], 14 disease-associated strains from dogs[51] and 521 strains from healthy cattle[52]. The APEC-associated genetic variants also occurred in humans and other

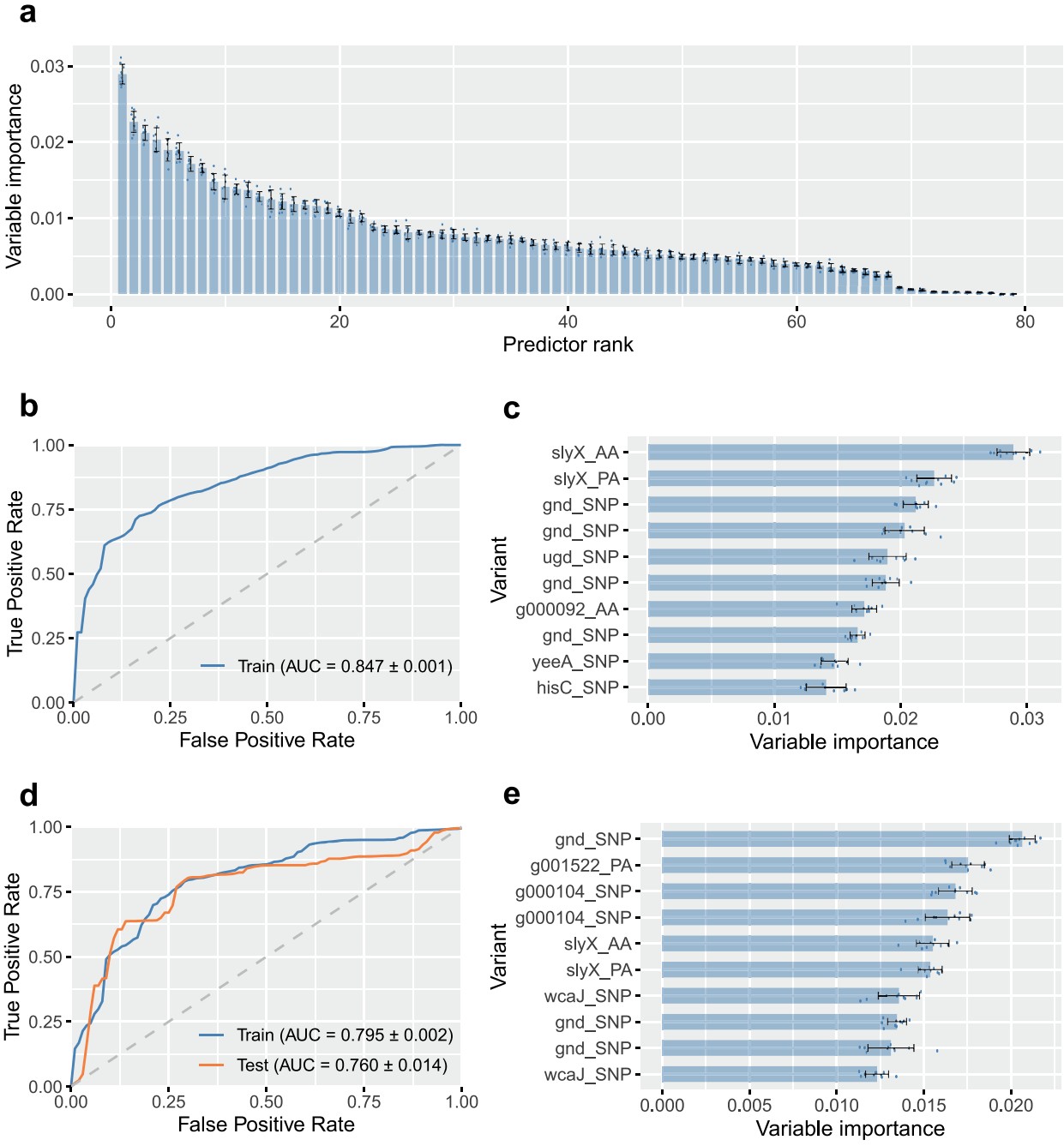

**Fig. 5 Identification of predictive genotypes for pathogenicity in avian *E. coli* using random forest (RF) models. a** The importance of the predictors derived from the four GWAS using the primary classifier model (trained using data from the four lineages A, B1, B2, ST-117); **b** Receiver operating characteristic (ROC) curve showing the overall performance of the primary classifier model; **c** Importance of the top 10 predictors in the primary classifier model; **d** ROC curve showing the overall performance of the follow-up classifier model (trained using data from the four phylogroups A, B1, B2 and predicting in ST-117); **e** The importance of the top 10 predictors in the follow-up classifier model. Data are presented as mean values ± SD from *n* = 10 repeated analyses. Source data are provided as a Source Data file.

animals implying that host species does not constitute a complete gene-pool barrier between the niches. A Kruskal–Wallis test identified significant differences in the prevalence of genetic markers in human ExPEC compared to the ECOR collection (*p*-value < 0.0001) and healthy bovine isolates (*p*-value < 0.0001), indicating that APEC-associated genetic variants can be found in other animals but overall they are significantly less abundant in human ExPEC strains (Supplementary Fig. 5b). However, ana-lysing the individual prevalence of specific APEC-associated genetic variants in each of these additional *E. coli* groups

(Supplementary Data 6) revealed that many of the APEC-associated elements were also common among human ExPEC (Supplementary Fig. 5c/Supplementary Data 6). This may suggest shared adaptations to establishing extraintestinal infection in both avian and human hosts[13].

Additionally, we investigated the prevalence of the same genetic variants in isolates from different infection sites within our dataset (Supplementary Fig. 5a/Supplementary Data 7). A Kruskal–Wallis test revealed significant differences between asymptomatic carriage and bone marrow samples

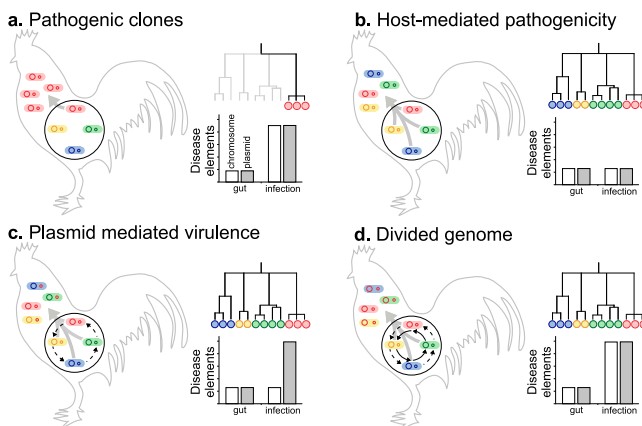

**Fig. 6 Evolutionary scenarios for APEC infection and predicted variation in isolate phylogenies and disease-associated elements.** Panels summarise models for the spread of *E. coli* from the primary commensal gut niche (black circle) to extraintestinal tissue, and the effect on the population of *E. coli* clones (blue, red, green and yellow circles) and their genomes (internal circles) which may be enriched for putative pathogenicity-associated genes (red). Conceptual genealogies are given for isolates sampled from extraintestinal disease sites and the estimated prevalence of chromosomal (white) and plasmid (grey) disease determinants in the genome of isolates from the gut and systemic infection are shown. Evolutionary scenarios include: **a** extraintestinal spread of pathogenic clones with genomes enriched for disease elements, seen as one (or few) lineages on the tree; **b** host-mediated pathogenicity in which multiple diverse clones spread systemically as a result of host factors, irrespective of disease elements (no significant difference); **c** plasmids transfer between lineages (dashed black arrow) and clones harbouring plasmids spread extraintestinally; **d** Horizontal gene transfer reassorts plasmid and chromosomal disease elements into multiple genomic backgrounds leading to the emergence of multiple APEC clones.

(*p*-value < 0.0001), between asymptomatic carriage and liver isolates (*p*-value = 0.0003) and between bone marrow and heart isolates (*p*-value = 0.009). While prevalence is distinct from GWAS-association, this suggests differences between infection types and provides evidence that the elements that underly pathogenicity may vary with different pathologies.

## Discussion

*E. coli* are simultaneously ubiquitous in healthy animal guts and a major cause of diverse intestinal and extraintestinal infections. Clearly, interpreting these contrasting lifestyles requires an understanding of the factors that promote pathogenicity in this, typically commensal, bacterium. By definition, extraintestinal disease requires migration from the gut and proliferation within the pathogenic niche. While host factors and dysbiosis may be important for this[27], disease also depends on the ability of the invasive strains to colonise a new habitat where the conditions are different.

Sampling *E. coli* from both healthy chicken guts and infected systemic tissues (APEC) allowed characterisation of the genotypes and gene pools associated with different sites, and health states. It is therefore possible to consider different evolutionary scenarios for the migration of pathogenic strains from the commensal gut niche and proliferation in extraintestinal tissues (Fig. 6). First, the emergence of dominant pathogenic APEC clones from the background gut population. Second, host-mediated pathogenicity, where all poultry-associated *E. coli* are able to cause extraintestinal infection. Third, plasmid-mediated virulence in which multiple lineages proliferate extraintestinally as a result of the acquisition of specific plasmid-borne virulence genes. Finally, a

divided genome scenario[53], where HGT introduces disease-associated chromosomal and plasmid genes into multiple genetic backgrounds allowing colonisation of the extraintestinal niche.

It is known that certain *E. coli* lineages are predisposed to extraintestinal pathogenicity, as evidenced by the global spread of pandemic ExPEC clones[5]. In a simple infection model, where only dominant clones can cause disease, all isolates recovered from infected tissues will belong to discrete clusters of genetically related pathogenicity-associated strains (Fig. 6a). This was not observed among APEC isolates. In fact, disease isolates were distributed across the phylogeny within all eight previously described phylogroups (Fig. 3a). One interpretation of this genetic structuring is that given particular host factors, such as gut perturbation or dysbiosis, all *E. coli* lineages are equally able to cause disease by mass action rather than specific pathogenicity[5] (Fig. 6a). If this were the case, then genome analyses would not identify enrichment of pathogenicity-associated elements within the genome of disease strains. However, the GWAS identified numerous pathogenicity-associated elements which mapped to genes known to be associated with pathogenicity. This is consistent with enrichment for sequence that encodes traits associated with pathogenicity.

Plasmid carriage is an important factor in the emergence of *E. coli* pathotypes[39]. The mobility of these elements makes them ideal candidates for spreading pathogenicity genes among APEC, potentially conferring multiple virulence phenotypes in a single evolutionary step. This could explain the emergence of pathogenic strains from divergent genetic backgrounds (Fig. 6c). However, our population-scale analysis revealed that known APEC plasmid genes were no more abundant in disease compared to commensal isolates from poultry (Supplementary Fig. 2). Furthermore, rather than containing a discrete compendium of genes associated with a given plasmid, genes were present at varying frequencies suggesting a mosaic of putative plasmid elements found in different combinations in the genome (Fig. 2 and Supplementary Fig. 2). While these findings are inconsistent with a simple model of plasmid-mediated virulence (Fig. 6c), the ubiquity of plasmid genes implies a role in poultry adaptation or the emergence of APEC as described in various studies[32,40,54].

Investigating the genetic basis of pathogenicity beyond the role of plasmids requires consideration of the putative function of all core and accessory genes. Pathogenicity is multifactorial and there is evidence that different traits can contribute in different *E. coli* lineages (Supplementary Fig. 3). Population-wide genomic screening and analysis approaches, such as GWAS and machine learning, deliver a deluge of potentially useful information describing the genetic basis of complex traits. If correctly integrated with laboratory microbiology, this can underpin rigorous confirmatory tests of gene function that satisfy Molecular Koch's postulates[55]. While functional genomic genotype–phenotype maps are required for a full understanding of pathogenicity, evidence for common APEC disease determinants comes from genes that are enriched in APEC strains from all *E. coli* phylogroups – consistent with convergent evolution of pathogenicity traits. Species-wide pathogenicity-associated genes included those associated with generic Gram-negative virulence factors, such as O-antigen chain length regulation (*wzzB*[56]) and a host killing toxins (*hokA* and *hokC*[57]), as well as known avian virulence factors including outer membrane proteins (*ompT*[58,59]), pilus chaperones (*papD* and *fimD*[60–62]), cell envelope integrity (*wcaJ*), and general secretory pathways (*gspO*) responsible for transport of toxins from the periplasm to the extracellular medium[63]. Antimicrobial resistance (*YeeO*[64] and *evgS*[65]) and central metabolism (*gnd*[47], *gltS*[66] and *hisBCDGH*[67]) genes also contained pathogenicity-associated elements, potentially conferring a survival advantage in the stress conditions of the infection niche[68].

Bacterial GWAS relies on whole-genome sequencing rather that standardised genotyping arrays and therefore typically has lower sample size than human GWAS (often >100,000 samples[69]). This can impact on the statistical ability to detect rare associated variants, but also on the ability to generalise results outside of the sampled dataset. Because of the lack of standardisation of bacterial GWAS results[70], and the scarcity of other comparable bacterial GWAS results, replication meta-analyses can be extremely challenging. Nevertheless, as for human GWAS, robustly associated variants in one study can always potentially highlight important functions and mechanisms associated with a trait[71,72], here pathogenicity in APEC. Therefore, any comparison with previously described virulence determinants can provide useful confirmatory evidence of a potential role for APEC associated genetic variation.

Cross-referencing with a recent authoritative review[73], identified GWAS associations in genes that have been linked to pathogenicity in other studies. For example, *eae* - an attaching and effacing gene that encodes intimin, and *ompT* - that encodes a protease able to cleave colicin. However, there was little overlap with some other studies[30]. It is inevitable that the findings of our population-wide approach will not exactly match data from existing microbiology studies for three important reasons. First, pathogenicity-associated elements are ranked based upon a significance score. With experimental design targeting ubiquitous APEC genomic signatures, this will inevitably flag pathogenicity elements found in multiple lineages and pathologies. Therefore, variation associated with a specific infection type may have lower significance. For example, while the *hokA* and *hokC* genes encoding components of toxin-antitoxin systems are not APEC virulence determinants in the strict sense, they may indicate the importance of wide-spread mobile genetic elements that are linked to virulence genes that vary by infection type. Second, many of the most significant hits were SNPs in core genes that may be linked to other genes (Supplementary Data 3). For example, while the chaperone-encoding genes *papD* and *fimD* are among the pathogenicity-associated genes, other P and type 1 fimbrial determinants are not, despite their likely role in virulence[74]. The reason for this is that the variation within the genes encoding fimbrial determinants does not segregate as strongly as the chaperone genes based on the binary pathogenicity phenotype in this study. This suggests that multiple homologous sequence variations within these genes can underlie pathogenicity when particular SNPs are present in the chaperone genes. Finally, factors promoting host colonisation, such as adhesion, can benefit commensal strains as well as representing a step towards bacterial pathogenicity in ExPEC[75–77]. In this case, while there may be a specific role in extraintestinal spread, the underlying genomic signature may not achieve statistical significance. For these reasons, the population-wide GWAS approach can be considered a platform on which to develop further functional genomic characterisation rather than a definitive road-map to pathogenicity, highlighting segregating variation in genes, such as *fimD*, that are linked to operons or pathways that are known to relate to *E. coli* virulence[61,73,74].

The enrichment of putative virulence determinants among disease isolates suggests that pathogenic strains are a subset of the commensal gut population that possess genetic elements that may promote migration and colonisation of extraintestinal sites (Fig. 6d). This presents something of a theoretical conundrum. Specifically, within the competitive milieu of the gut microbiota (primary niche), genes that are only beneficial in the secondary (extraintestinal) niche will impose a fitness cost on the bacterium. Therefore, strains with many pathogenicity adaptations will be less competitive than those with few. A clue to explaining this comes from the observation that 74% of the pathogenicity-associated accessory genes were putatively of plasmid or phage origin and are therefore mobile among lineages. HGT is known to be a major force in *E. coli* evolution[78,79], with the acquisition of genes through recombination potentially conferring adaptations associated with pathogenicity[80]. Therefore, virulence genes that are maintained at low frequencies in different lineages in the primary niche can be reassorted and come together in a common genetic background, potentially allowing successful extraintestinal colonisation.

Evidence of the importance of HGT in APEC comes from the divergent position of strains across the population phylogeny (Fig. 3a), as well as the lower mean consistency index of individual pathogenicity-associated gene trees compared with core genes. This suggests homoplasy and the horizontal spread of adaptive genes through the population, consistent with a gene-specific selective sweep, or divided genome, model of bacterial evolution[53,81,82]. In this scenario, as migration from the gut occurs, HGT will increase the rate at which positively selected genes sweep through the invasive population. The speed and efficiency of adaptation are therefore increased by recombination combining multiple selected plasmid and chromosomal genes into a common genomic background (Fig. 6d). For diverse commensal *E. coli* populations this can potentially promote the emergence of pathogens at the boundary between commensal and extraintesinal niches and rapid adaptation to life in the extraintestinal environment. Furthermore, studies of in vivo bacterial populations have demonstrated recombination among strains within the gut of humans and chickens[83,84]. Knowing that *E. coli* can be found at high concentrations in chickens (mean $\log_{10}E.$ *coli* of 4.15 colony forming units per ml of faeces[85]), with an estimated doubling time of around 3 to 15 h in natural populations[86,87], it is also possible that in chronic APEC infections, lasting more than 2 days from experimental infection to the death of the bird[88], virulence determinants could accumulate and reassort among strains.

In an animal health setting, early identification of pathogens has great potential to improve livestock welfare and reduce economic losses resulting from disease. It is evident that targeting individual clones based upon traditional molecular typing methods[89] has limitations because the putative virulence determinants are mobile between lineages. It follows, therefore, that quantifying pathogenicity-associated genes may allow the identification of carriage strains that pose a disease risk. However, this is complicated by two factors. First, there is evidence of lineage-specific and species-wide pathogenicity-associated genes so simple diagnosis may be challenging. Second, extraintestinal spread and systemic infection involves numerous colonisation and virulence factors that may be associated with progression of different types of infection, such as common respiratory tract infections and septicaemia. Statistically significant GWAS correlation with different infection types would require larger numbers of samples from each extraintestinal source but varying prevalence of pathogenicity-associated elements among isolation sources was consistent with multiple pathways to infection.

To achieve more accurate risk prediction of this complex disease, we developed a Random Forest machine learning approach to quantify the power of different combinations of pathogenicity-associated elements (classifiers) to predict the source of isolates (carriage/disease) from their genome. A simple analysis, in which all isolates were used to train the model, achieved a classification accuracy of 76.9%. Among the classifiers that provided the most accurate prediction were elements in a gene associated with polymyxin resistance (*ugd*[90]) and a gene participating in the oxidative pentose phosphate metabolic pathway (*gnd*[47,91]) located in the highly recombinant *rfb* region associated with the avoidance of host defence systems[92]. This may be explained by the use

of polymyxin in the treatment of colibacillosis in poultry production[93] and HGT among strains, and provides clues about the functional basis of the pathogenicity-associated elements. However, as APEC emerges in multiple *E. coli* lineages, generalising the risk prediction method required that the training and test datasets were phylogenetically distinct. Focussing on the ST-117 lineage, that is thought to be an emergent bird pathogen[49] achieved a risk prediction accuracy of 72.7% when the model was trained on phylogroups A, B1 and B2. This suggests that relatively few 'global' pathogenicity markers may provide a basis for risk prediction and that model training on ever larger reference genome datasets may have potential for early identification emergent APEC in healthy flocks to inform targeted interventions.

Pathogens remain a major threat to sustainable livestock production. Control of highly pathogenic avian influenza is typically achieved through early diagnosis, flock isolation and bird culling. However, these measures are not applicable for some common diseases, such as colibacillosis, because *E. coli* are found in all chicken guts and a full understanding of the factors responsible for the emergence of APEC has remained elusive. Population-scale comparative genomic analyses take us a step closer to characterising the differences between commensal gut *E. coli* and APEC that have acquired genetic elements that promote extraintestinal infection. Untangling the web of interacting genes that underly pathogenicity is extremely difficult without full functional genomic characterisation and it is inevitable that some lineage- or pathology-specific genetic variation is missed when targeting species-wide markers of infection. However, there is clear utility for the development of APEC molecular diagnostics and targeted antibiotic therapy - where pathogens have a different resistance profile[94,95]. Among the most promising applications are those involving the development of guided antimicrobials that can selectively target a gene, cellular process, or strain of choice. Where the pathogen sequence is known, nucleic acid-based antibacterials, peptides, bacteriophage therapies, bacteriocins, and anti-virulence compounds may be able to exclusively target disease-causing bacteria[94,96]. For example, CRISPR-Cas technology can be used for sequence-specific bacterial killing through guided nucleases that recognise known DNA sequences[97,98]. Despite the complexity of pathogenicity phenotypes, and the mobility of the underlying genes, these techniques show considerable potential. With further functional genomic validation, a firm understanding of the genetics of pathogenicity could pave the way to early diagnosis of risk, more effective control and improved animal welfare.

## Methods

**Bacterial sampling.** The isolate collection comprised 568 avian *E. coli* isolates sampled from a variety of sources (Supplementary Data 1). These included 152 previously published isolates[50,58,99], 414 isolates sequenced as part of this study and the reference strains APECO1[100] and APECO78[101]. In total, 482 isolates were sampled at slaughter from broiler chicken (*Gallus gallus domesticus*), 44 isolates were from commercial turkey (*Meleagris sp.*), 12 isolates were from avian wildfowl and 5 isolates were from gulls. A total of 234 faecal isolates were from asymptomatic carriage, sampled post mortem from the gut when no symptoms could be observed in the bird, and 309 isolates were considered disease-associated APEC based on their extraintestinal site of isolation, and/or symptoms in the bird at autopsy. Specifically, out of the disease-associated isolates 35 were isolated from the bone marrow, 70 from the liver (perihepatitis), 23 from the heart (pericarditis), 40 from the peritoneum (peritonitis), 26 from blood (septicaemia), 19 from yolk sac infections and 96 from various other infection sites. Finally, 25 isolates were not phenotypically characterised as they were isolated from the poultry farm environment. Isolates from the poultry farm environment were only used for the phylogenetic and pangenome analysis of this study.

**Genomic DNA extraction, sequencing and archiving.** DNA was extracted using the QIAamp DNA Mini Kit (QIAGEN; cat. number: 51306), using

manufacturer's instructions. DNA was quantified using a Nanodrop spectrophotometer, as well as the Quant-iT DNA Assay Kit (Life Technologies, Paisley, UK). High-throughput genome sequencing was performed on a MiSeq (Illumina, San Diego, CA, USA), using the Nextera XT Library Preparation Kit with standard protocols. Libraries were sequenced using $2 \times 250$ bp paired end v3 reagent kit (Illumina), following manufacturer's protocols. Short read paired-end data was assembled using the de novo assembly algorithm, SPAdes (version 3.10.0)[102]. The average number of contigs was 336 (range: 11–1373) for an average total assembled sequence size of 5.16 Mbp (range: 4.42–5.79). All 414 genomes sequenced in this study have been deposited in GenBank, associated with BioProject PRJNA592536. Accession numbers for all genomes, including those previously sequenced can be found in Supplementary Data 1. Genome assemblies for the entire collection 568 can be downloaded from figshare: https://doi.org/10.6084/m9.figshare.12011811. An overview of the assembly information is provided on Supplementary Data 4.

**Core and accessory genome characterisation.** All unique genes present in at least one isolate (the pangenome) were identified by automated annotation using PROKKA (version 1.13)[103] and PIRATE[104] - a pangenomics tool which allows for orthologue gene clustering in divergent bacterial species. We defined genes in PIRATE using a wide range of amino acid percentage identity thresholds for Markov Cluster algorithm (MCL) clustering (45, 50, 60, 70, 80, 90, 95, 98). Additional APEC reference genomes and APEC associated plasmids were included in the pangenome to maintain locus nomenclature and identify plasmid carriage, including APEC_O1 (accession: GCA_000014845.1)[6], APEC_O78 (accession: GCF_000332755.1)[26], APEC_IMT5155 (accession: GCF_000813165.1)[32] and *E. coli* 789 (accession: GCF_000819645.1)[25]. Genes in the pangenome were ordered initially using the APEC_O1 reference followed by the order defined in PIRATE based on gene synteny and frequency. To perform core and accessory pangenome variation analyses a matrix was produced summarising the presence/absence and allelic diversity of every gene in the pangenome list[105,106]. Core genes were defined as present in 95% of the genomes and accessory genes were present in at least one isolate. The number of genes detected in each strain was calculated by PIRATE and can be found in Supplementary Data 4. Using this approach, the amount of coding sequences detected per strain were not significantly affected by the quality of the assembled genomes (Supplementary Fig. 4).

**Phylogenetic analysis.** Phylogenies were constructed by mapping pseudoreads simulated from assembled genomes to the *E. coli*_O157 RefSeq reference genome (accession: NZ_CP015831.1; 5,831,209 bp)[107] using snippy[108].

This well-characterised closed reference genome, from Phylogroup E, was absent from the collection under investigation. This minimised and standardised bias caused by reference strain selection[109]. As all isolates were mapped to a single reference, this also allowed for a tree to be constructed from all isolates and comparison of genomic/alignment regions used for tree building during the recombination analysis. Other references, including APEC_01, were included in the pan genome analysis in order to provide points of reference for comparison of genes to well-characterised reference genomes. Pseudo-reads were created as a part of the SNIPPY pipeline used for variant calling. Contigs passed to SNIPPY were split into 250 bp single-end read pairs at a simulated ~20x coverage of the reference genome. These pseudo-reads were mapped against the reference genome O157 in the same manner as trimmed reads to retain order and ensure all data were comparable. Maximum-likelihood phylogenies were constructed separately for phylogroups A, B1, B2 and ST117, and the complete collection using a GTR + I + G substitution model and ultra-fast bootstrapping (1000 bootstraps) implemented in IQtree (version 1.6.8)[110] and visualised on Microreact[111]. Putative recombination sites were inferred using ClonalFrameML[112] and masked using cfml-maskrc (https://github.com/kwongj/cfml-maskrc). Recombination-masked alignments were used to build midpoint rooted trees, which were used in treeWAS to weight associations, accounting for lineage effects[113] (Supplementary Fig. 3).

**Pangenome-wide association study of infection-associated genes.** Four GWAS analyses were performed to identify pathogenicity-associated core and accessory genome variation in the most common *E. coli* lineages, specifically phylogroups A ($n = 71$), B1 ($n = 85$) and B2 ($n = 152$), and the ST-117 lineage ($n = 220$). The remaining phylogroups contained too few isolates for reliable GWAS analysis. GWAS were performed using treeWAS[35] incorporating core and accessory genome variation identified by PIRATE[104]. We used the core and accessory genes to investigate associations representing segregation (disease vs. carriage isolates) of: (i) core SNPs; (ii) accessory gene presence/absence; (iii) gene fissions/fusions; (iv) gene duplications; (v) accessory gene alleles. Fission/fusion genes are identified by PIRATE[104] as genes which, due to nonsense mutation or frameshifts, comprise a single ORF in at least one isolate but two or more distinct ORFs in other isolates within the collection. Variants with an allele frequency <0.01 were excluded from the GWAS. The treeWAS algorithm performs three statistical association tests to calculate terminal, simultaneous and subsequent association scores. Infection-associated variants ($p$-value < 0.05) in any of these three tests were further

investigated[114], and elements that were significantly associated in all four GWAS experiments (*p*-value < 0.01) were identified[35].

**Quantitative analysis of putative plasmid genes**. Plasmid-associated genes are known to be important in APEC virulence[31–33]. We therefore identified putative plasmid replicon sequences in all the isolates using PlasmidFinder 2.1[115] (Supplementary Data 5). However, these genes can be located on a plasmid or the bacterial chromosome, for example in genomic islands. This means that analyses that focus on entire plasmids may miss the mosaic of possible plasmid and chromosomal genes that are known to recombine extensively in avian *E. coli*[116]. Confirmation of the plasmidic context of specific genes may require that analyses are performed separately on miniprep plasmid extractions or the use of long read sequencing[117]. However, the prevalence of putative APEC plasmid genes among isolates from carriage and invasive disease can be determined using the gene-by-gene approach[37,38,118] employed here. Additionally, annotated gene lists were obtained for the following plasmids: pAPEC-O1-R[119]; pAPEC-O2-R[120]; pAPEC-O2-211A-ColV[121]; pAPEC-O1-ColBM[54]; p1ColV5155[122]; pAPEC-O2-ColV[10]. To detect additional genes of plasmid origin among the infection-associated genes we used the PlasmidSeeker database (latest update on 12/07/2017)[123]. Specifically, all plasmid genes were annotated for putative function using the PlasmidSeeker database and PROKKA 1.13[103] to create a database compatible with Abricate 0.8.13[124], which was interrogated in relation to the list of the GWAS infection-associated genes. Putative plasmid genes were identified in all isolate genomes as a BLAST match of >70% over >50% of the gene length. For every plasmid, the number of genes present in each isolate was quantified and the distribution among asymptomatic and disease isolates was compared using an unpaired *t*-test.

**Horizontal gene transfer among infection-associated genes**. Population genetic analyses was undertaken to compare molecular variation among the 143 genes that contained infection-associated elements and the core genome of the dataset in this study. The number of alleles per locus was determined using a whole-genome MLST approach[37], and the consistency of the phylogenetic trees to patterns of variation in sequence alignments for each gene was calculated[125,126] to infer the minimum amount of homoplasy in infection-associated and core genome genes. We defined the number of alleles per locus as the number of unique alleles per gene divided by the total number of isolates. Consistency indices (CI) for each single-gene alignment of 143 infection-associated genes to a phylogeny constructed using an alignment of 3094 core genes shared by 570 isolates, were calculated using the CI function of the R *Phangorn* package[127]. The average CI of these shared genes was compared with the CI of the genes containing pathogenicity-associated elements.

**Risk calculation**. Invasive infections caused by APEC are associated with multiple factors and large numbers of disease-associated elements within the bacterial genome. Therefore, while GWAS improves understanding of genome evolution and pathogen emergence, translating these findings for practical risk prediction models can be challenging. To achieve this, we trained a Random Forest (RF) classifier[128] using the GWAS output. This allowed us to capture the potentially complex, non-linear relationship between presence/absence patterns of disease-associated elements and phenotype, and rank the features according to their power to predict the isolate source (invasive disease vs. carriage). Analyses were conducted in R[129] using RandomForest[130], ROCR[131] and ggplot2[132] software using the 510 isolates (291 invasive disease, 219 carriage) used in the GWAS with 79 binary presence/absence species-wide predictors used to train the RF model (Supplementary Data 3). In separate RF analyses, the classifiers were trained with (i) all 510 isolates and (ii) 294 isolates from phylogroups A (*n* = 70), B1 (*n* = 80) and B2 (*n* = 144) with ST-117 (*n* = 216) isolates as a test set. Based on the training data, a RF model with 1000 trees estimated the importance of the predictors with model criteria for feature selection. To estimate the out-of-sample accuracy of the model within its training domain (as specified by the phylogroups), the out-of-bag (OOB) predictions were used. In addition, in the second analysis the model was evaluated on the test set, which contained isolates from outside the training domain of the RF model. The predictive performance of the models was evaluated by predictive accuracy and area under the receiver operating characteristic (ROC) curve. Each analysis was repeated ten times and the reported results are the average over the ten independent runs. To investigate the prevalence of the APEC-associated genetic variants used in the RF model, in isolates sampled from other host niches we used four additional *E. coli* collections. Specifically, we analysed the *E. coli* reference collection (ECOR)[42], 175 human ExPEC strains[28], 14 disease-associated strains from dogs[51] and 521 strains from healthy cattle[52]. Published genomes were uploaded to BIGSdb[133] and presence-absence matrices were created for the APEC-associated SNPs and accessory genes (excluding fission/fusion and accessory gene alleles). The number and prevalence (%) of APEC-associated variants were compared for the four additional groups (Supplementary Fig. 5).

**Reporting summary**. Further information on research design is available in the Nature Research Reporting Summary linked to this article.

## Data availability

Short-read sequence data for all isolates sequenced in this study are deposited in the sequence read archive (SRA) and can be found associated with BioProject #PRNJA592536. Assembled genomes are also available on Figshare (https://doi.org/10.6084/m9.figshare.12011811). NCBI genome accession numbers for isolates in the validation dataset are included in Supplementary Data 1. Source data are provided for this paper.

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

## Acknowledgements
This work was supported by Wellcome Trust grants 088786/C/09/Z and Medical Research Council (MRC) grants MR/M501608/1 and MR/L015080/1 awarded to S.K.S.

## Author contributions
S.K.S., L.M. and G.M. conceived and designed the study. L.M., G.M., B.P., S.K.S., S.M., J.C., T.S.W., and L.K.W. carried out Laboratory work. B.P., K.A.J., and S.K.S. supported data archiving. L.M., G.M., S.B, K.Y., E.M., J.P. and J.C. analysed the data. E.J.F., S.C.B., M.D.H., J.P., K.K., N.J.W. and J.C. contributed to data interpretation. S.K.S. L.M. and G.M. wrote the paper.

## Competing interests
The authors declare no competing interests.
