## [Peer Review File · Nature Communications]

REVIEWER COMMENTS

Reviewer #1 (Remarks to the Author):

Mageiros et al. presents their results on large-scale comparative genomics analysis of 568 avian *E. coli* genomes to investigate the genetic basis of APEC pathogenicity. Their results indicate pathogenicity emerging from multiple genetic backgrounds, presence of specific pathogenicity-associated genes, frequent recombination of pathogenicity-associated genes and predictive power to identify pathogenicity risk markers. This is a well-written and comprehensive manuscript that presents important insights on avian *E. coli* that is poorly examined compared to human-associated *E. coli*. Below are my comments.

Major comments:

1. Table S4: The number of contigs per genome range from 1-1373, with 109 genomes having >500 contigs, and N50 values that are as low as 6338bp and 198 genomes with <50,000bp. Assembly metrics can have a tremendous impact on any pan-genome analyses, including uncertainty in how orthologous genes are defined and missing genetic segments/sites, contamination and completeness of assemblies. How did the authors ensure that those genomes with very high number of contigs and those with very low N50 values have not significantly impact their results? How are sequencing and assembly qualities measured? Please also comment on whether any filtering of genomes used for downstream analyses was done (in Methods) and the limitations this may present to your analyses (in Discussion).

2. Discussion, Lines 353-354: What are examples of targeted interventions that can be used against colibacillosis in poultry production? While it is interesting that pathogenicity-associated elements have been identified in APEC, it is unclear what kind of targeted interventions can be developed and applied to poultry health.

Minor comments:

1. Introduction, Line 94: "...about [add "the"] extent to which..."
2. Results, Line 143: plasmid-borne not plasmid born
3. Results, Line 190 and Methods, Line 432: Need to describe how fission/fusion genes were identified
4. Results, Line 195-196: "... the importance [add "of"] potentially mobile elements..." Throughout the manuscript: "extraintestinal versus extra-intestinal" Just choose one and be consistent in using it. Same as in "disease associated versus disease-associated" and "pangenome versus pan-genome"
5. Methods, Line 468: Remove comma in "...to capture the, potentially complex..."
6. Results, Line 216: Need to spell out ROC the first time it is mentioned. ROC was spelled out in the Methods.
7. Figure 6 Legend, Line 551: Remove "and" in "...irrespective of and disease elements..."
8. Figure S2 Legend, Line 565: "The frequency [add "of"] known APEC-associated..."

Reviewer #2 (Remarks to the Author):

In this study genome sequences of 568 avian *E. coli* isolates were analyzed. This collection includes sequences from disease-associated strains from various sites of isolation as well as genomes of asymptomatic fecal colonizers. Additionally, also a small number (n=18) genomes represent isolates from the surrounding environment. The draft genome sequences have been used to calculate pseudoread-dependent core genome-based phylogenies for phylogroups A, B1, B2 and sequence type 117. The authors applied pan-genome-wide association study to identify infection-associated genes using treeWAS. As a result, this data set confirmed previously published findings that APEC can be allocated to a broad variety of phylogenetic lineages. This study presents a first initial try to identify disease-associated alleles by GWAS leading to large groups of "infection-associated genes" in different phylogroups or lineages. Many of these genes were phylogroup or lineage-specific. 143 genes appeared to be disease-associated in all phylogroups.

Interestingly, a high percentage of poultry isolates of this data set belonged to ST117. The genome sequences were also searched for putative plasmid genes using BLAST searches. Based on gene-by-gene comparisons, it seems as if the number of putative plasmid genes was higher in asymptomatic colonizers than in disease-associated isolates. Both isolate groups could not be distinguished based on the presence/absence of certain plasmid genes. The authors also investigated recombination between the 143 "infection-associated" genes. The comparison of allelic variation in the core genome and in the 143 "disease-associated" genes may indicate that the "disease-associated" genes frequently undergo recombination events and exchange by horizontal gene transfer. Finally, machine learning was applied to correlate the presence/absence of 79 disease-associated genes and disease phenotype. Based on this analysis, ten predictors were determined, which allow classification of strains with ~77 % accuracy. SNPs in *gnd* and *slyX* were identified to exhibit considerable predictive potential that may be useful for the development of biomarkers. The study represents a valuable comparative genomic analysis aiming at the identification of pathogenicity markers of APEC. A major advantage of the study is the inclusion of disease-associated isolates from different body (infection) sites and of asymptomatic faecal isolates in the analysis. The design of the study is convincing with regard to the analysis of the phylogenetic diversity of avian *E. coli* and the prediction of genomic differences between disease-associated and asymptomatic fecal colonizers. However, against the background of the title of the manuscript, the data presented are not convincing in terms of statements on "evolution and emergence" of pathogenicity.

The authors confirmed once more that APEC can derive from several major phylogenetic lineages. The description of 143 disease-associated genes that appear not to be lineage-specific points towards promising candidates. In order to confirm this assumption, which is so far only based on comparative genomic prediction, and to support the suggestion that these "markers" could indeed promote risk prediction, experimental evidence for their role in pathogenicity is required. It should be experimentally tested and explained how, e.g. certain allelic variants of house keeping genes or O antigen biosynthesis genes contribute to increased pathogenicity.

It is understandable that in the search for pathogenicity relevant markers of APEC only avian isolates were analyzed and it is greatly appreciated that the authors included a large and diverse group of strains into their study. However, in the context of analyzing potential disease-associated markers, it would be important to know that the predicted markers do not also occur in other *E. coli* pathotypes. This aspect is missing in the study so far. It will be important to show that the predictors that result from the GWAS and machine learning approaches are actually discriminatory for APEC. Can such alleles also be found, for example, in human strains or in extraintestinal pathogenic *E. coli* isolates from cats and dogs?

The authors did not find many "typical" *E. coli* virulence genes, such as adhesins, T3SS, T5SS, T6SS secretion systems, etc., but many house keeping and metabolic genes. As the authors themselves write in the introduction, *E. coli* infections in poultry are very diverse and result from different infection routes. Accordingly, aerosacculitis develops and manifests differently from omphalitis or salpingitis. The route of infection could indicate a dependence on certain virulence factors that are required for this. The selectivity of the analyses presented in the manuscript could suffer from this in the same way as from the fact described by the authors that the pathogenicity potential of asymptomatic colonizers of the intestinal tract cannot be predicted exactly. To increase the selectivity, GWAS analyses should not only be performed with different phylogroups, but also with isolate groups distinguished on the basis of the type of infection, as the mechanisms and routes of infection or colonization seem to play an important role.

Why has *E. coli* O157 been used as a reference for cgenome-based phylogeny of APEC?

Many plasmid-associated genes can also be located on the bacterial chromosome, e.g. in genomic islands (typical examples are siderophore determinants, e.g. coding for aerobactin or salmochelin, autotransporter genes, T3SS gene clusters). How can the authors distinguish between chromosomal and plasmid genes? BLAST searches alone are not really helpful. For a proper allocation of genes to either the chromosomal or to plasmid(s), their sequence context would have to be considered as well. Accordingly, the statements regarding plasmid genes should be cautiously interpreted. Wouldn't it be also helpful to assess the prevalence and diversity of plasmid

types in APEC? The authors should include pMLST and incompatibility group-typing of the plasmid-derived contigs in the genome sequences.

How relevant and meaningful are the statements in lines 287-299 with regard to the generated data and their interpretation regarding the ability of APEC to cause infection in contrast to asymptomatic colonizers? The *hokA* and *hokC* genes coding for components of toxin-antitoxin systems are certainly not APEC virulence determinants in the strict sense, but indicate the presence of plasmids or derived mobile genetic elements (on which possibly much more relevant genes could be located...). Why do the authors find only the chaperone-encoding genes *papD* and *fimD* among the virulence-associated genes and not the entire P or type 1 fimbrial determinants? Is the assembly of P- and type 1 fimbriae affected by these selected chaperone alleles? Which APEC toxins should be secreted by the general secretion pathway? For typical toxins of extraintestinal pathogenic *E. coli* (e.g. haemolysin and different cyclomodulins) the general secretion pathway does not play a mechanistic role. Evidence of a specific advantage from the presence of these gene variants in APEC is missing. The mutation rate and possible selection conditions have not been tested, which could provide alternative indications of possible allele variations independent of diseases association.

The suggestion that the presence of accessory disease-associated genes on mobile elements represents an evolutionary advantage, because it allows rapid accumulation and reassortment to suit changing environmental conditions is interesting. Do the authors know gene transfer rates in the intestinal tract or in extraintestinal Habitats? What is known about the time frame of extraintestinal infection in poultry to enable such processes? Shouldn't the authors assume chronic extraintestinal infections to allow for virulence determinants in APEC to accumulate and reassort to this extent?

Reviewer #3 (Remarks to the Author):

This study by Mageiros and colleagues examines the genetic elements that distinguish avian pathogenic *E. coli* (APEC) from commensal *E. coli* located in the avian gut. It also discusses the likely evolutionary processes that dominate the shift between the invasive and commensal states, arguing that horizontal gene transfer is the dominant mechanism. Finally, the study presents a random forest model for predicting pathogenicity based on a limited set of genetic markers.

The study is methodologically solid and the claims are well founded. It is not groundbreaking in terms of methodology. In fact, it is almost a step-by-step recreation of Méric and Mageiros et al, 2018 (<https://www.nature.com/articles/s41467-018-07368-7>), only with a different organism. APEC is nearly an ideal choice of organism for such a study though, as it is known to emerge after acquiring pathogenicity elements, is a high-impact disease on welfare, production and economy, and might be preventable if its emergence was better understood.

The central ideas of the paper are presented in a logical order and makes for an entertaining read. The conclusion that HGT is a more dominant mechanism for the emergence of APEC than plasmid transfer is original, and might be somewhat controversial, because the presence of various plasmids has been seen as the defining characteristic of APEC and a solid way to distinguish one from the other.

For me, the major weakness of the paper is that there is too little focus on putting its findings into context with existing literature. For another example, the paper presents a list of genes that were most highly associated with pathogenicity. (Highlights in the results section and full list in supplementary). This list has little overlap with "established" APEC virulence factors. For example, the paper cites Delicato et al (2003), which posits that *iutA*, *iss*, *cvaC*, *tsh*, *papC*, *papG* and *felA* are virulence genes. None of these are found in the results here and the results are not discussed. Similarly, Janssen et al (2001) is cited, but the virulence genes found in that paper is not put into context of the current manuscript (*iucD*, *tsh*, *fimC*, *fyuA* and *irp2*). A very recent resource is the textbook chapter on colibacillosis by Nolan et al 2020 (<https://doi.org/10.1002/9781119371199.ch18>), which lists a large number of virulence genes

ordered by different functions. Again, there is not much overlap between these virulence factors and the ones presented from the GWAS model of the current paper.

The manuscript presents a very interesting idea, that some strains can be "predisposed to pathogenicity". This is first mentioned in the introduction and then again at the end of the results and in the discussion. But if this is a goal of the paper, it is never fully achieved. It would be very useful if a set of sufficient (or necessary) conditions for pathogenicity could be found. (True, this would be a daunting task.) The manuscript mentions as complicating factors lineage-specific and tissue-specific virulence factors. I would have liked to see this idea explored further. For example, it is clear that the authors have information on the extraintestinal location of APEC isolates, but this information does not appear to have been included in the GWAS analysis?

As for lineage-specific virulence factors, that might indeed make the whole thing more complex. However, the authors' own RF model drops only about 3% from training on the entire set to training on A,B1 and B2, and testing on ST-117. As they correctly point out, that means that a limited set of global pathogenicity markers must be the driving forces. In the RF model, a majority of the predictors are core genome SNPs in hypothetical proteins. This makes it seem likely that rather than predicting APEC/commensalism from virulence factors, the model is probably predicting some kind of phylogenetic signal, or mutations that are hitchhiking with real virulence factors in HGT.

Below I have comments to specific parts of the manuscript:

- Availability of data: Looking at bioproject accession PRJNA592536 there seem to be missing some data. That accession only contains data from 259 isolates. The paper refers to 568 isolates, and from Supplementary table S1 it's clear that 414 are from "This study".

- Please double check that all the numbers in the manuscript are correct and correspond to those of the tables. For example, I have found the following inconsistencies:

- (1) - In line 158, 220 isolates are said to be ST-117. However, in the supplementary table, only 211 are listed.

- (2) - In the sampling section of the materials and methods, isolate numbers by host: $475 + 33 + 12 + 5 = 525$. But the total is supposed to be 568?

- (3) - In the same section, 243 were from asymptomatic, 307 were APEC. What about the final 18? Also, Suppl table 1 only lists 242 as asymptomatic.

- Results, line 178-182 - The number of discovered elements with $p < .05$ is very high - Over 43,000 for B2 alone! Over all the major phylogroups, the significant GWAS results links 5200 genes to invasiveness. This seems like an extraordinarily high number, seeing as the total pan-genome is around 15,000 genes for this collection. (Evident from suppl fig 1). Now, looking at figure 3 B, it appears as if the number of genes included in the analysis is roughly 10,500. Does this mean that half of all genes had some variant linked with pathogenicity? That seems rather extreme. How many of these are simply phylogenetic variants that predict a single invasive cluster?

- In the following sentence, 753 of these have $p < .01$, and exactly the same number are found to be lineage-specific? Is this a coincidence or is there a numbering error here?

- 143 genes are identified as significant in the GWAS model across all phylogroups. From these candidate genes, the list is trimmed to 79 predictor elements by selecting only those with complete(?) segregation between commensal and pathogenic. It's unclear to me why such a strict criteria was imposed, because it means that the predictor list does not contain any elements from many of the genes that were predicted in the GWAS to be most associated with disease. For example, the top GWAS hit, *yeeO*, has no associated elements in the predictor list.

- Lines 198-200 - 0.156 unique alleles per locus for the 143 pathogenicity-associated genes. - I would have liked to have an explanation in the methods section of how this metric is calculated because I don't understand it. How is there less than one unique allele per locus? How can any genes have zero number of unique alleles per locus? (Fig 4)

- Supplementary table S1 - It would be nice if the individual run accessions were listed for each isolate.
- Supplementary table S1 - Some metadata missing. I'm sure the authors know what year and country their own isolates hail from, and this should be entered in the table.
- Supplementary table S1 - Some hard to interpret metadata. What is "HTP"? What is "boot drag"?

- Phylogenetic analysis - Pseudoreads were generated from assembled genomes. However, there are no details on how the genomes were assembled initially nor how the pseudoreads were generated.

- Fig 3, panel B - The outside ring. What is the unit? Is it gene number or does it correspond to genome coordinates?

RESPONSE TO REVIEWERS

Reviewer #1 (Remarks to the Author):

This is a well-written and comprehensive manuscript that presents important insights on avian *E. coli* that is poorly examined compared to human-associated *E. coli*. Below are my comments.

We thank the reviewer for the positive comments and respond below.

Major comments:

1. Table S4: The number of contigs per genome range from 1-1373, with 109 genomes having >500 contigs, and N50 values that are as low as 6338bp and 198 genomes with <50,000bp. Assembly metrics can have a tremendous impact on any pan-genome analyses, including uncertainty in how orthologous genes are defined and missing genetic segments/sites, contamination and completeness of assemblies. How did the authors ensure that those genomes with very high number of contigs and those with very low N50 values have not significantly impact their results? How are sequencing and assembly qualities measured? Please also comment on whether any filtering of genomes used for downstream analyses was done (in Methods) and the limitations this may present to your analyses (in Discussion).

We welcome the reviewer's comment. More and more high quality 'closed' bacterial genomes are becoming available for bacteria. However, for the time being the added information from sequencing more isolates often outweighs the benefit of very high-quality genomes, particularly in understudied species/strains (eg. APEC) where the open pan-genome continues to reveal extraordinary variation in putative gene function. This led us to follow focus on sequencing large numbers of isolates and to use a concatenated gene-by-gene alignment approach consistent with many bacterial genomics studies. This approach reduces dependence upon very high-quality assemblies to robustly determine sequence, as evidenced by the comparable amount of coding sequences obtained for all isolates irrespective of the number of contigs. To satisfy potential concerns we have updated assembly metrics in Table S4, added supplementary figure S4, and described these in the manuscript text (lines 490 - 492 and 681 - 687).

2. Discussion, Lines 353-354: What are examples of targeted interventions that can be used against colibacillosis in poultry production? While it is interesting that pathogenicity-associated elements have been identified in APEC, it is unclear what kind of targeted interventions can be developed and applied to poultry health.

*Poor understanding of the genes that differentiate APEC from commensal *E. coli* has made it difficult to develop accurate diagnostics and targeted interventions. Currently, most detection of APEC infection in chicken is post-symptomatic. Therefore, entire flocks can sometimes be infected before any action is taken. However, more precise characterization of pathogenicity genes can provide a platform for the development of molecular diagnostics, informed antibiotic therapy and guided antimicrobials that can exclusively target disease-causing bacteria. Spurred by the potential economic benefits, strain-specific killing in livestock is becoming a major development area for industry. For example, the authors are currently working with UK partners (<https://foliumscience.com/>) on a CRISPR-Cas delivery system targeting known APEC sequences in vivo. As suggested by the reviewer we have now included more information and references about the nature of potential targeted interventions for poultry health (lines 433 - 442).*

Minor comments:

1. Introduction, Line 94: "...about [add "the"] extent to which..."

This has been corrected.

2. Results, Line 143: plasmid-borne not plasmid born

This has been corrected throughout.

3. Results, Line 190 and Methods, Line 432: Need to describe how fission/fusion genes were identified
Fission/fusion genes are identified using PIRATE as genes which, due to nonsense mutations or frameshifts, comprise a single ORF in at least one isolate but two or more distinct ORFs in other isolates within the collection. Sequence homology is established using BLAST to identify ORFs with little to no overlap within a genome but which map to distinct sections of the longest representative ORF of a gene family (previously identified by PIRATE using BLAST-MCL). (lines 522 - 524).

4. Results, Line 195 -196: "... the importance [add "of"] potentially mobile elements..." Throughout the manuscript: "extraintestinal versus extra-intestinal" Just choose one and be consistent in using it. Same as in "disease associated versus disease-associated" and "pangenome versus pan-genome"

We now consistently use 'extraintestinal', 'disease-associated' and 'pangenome'.

5. Methods, Line 468: Remove comma in "...to capture the, potentially complex..."

This has been removed.

6. Results, Line 216: Need to spell out ROC the first time it is mentioned. ROC was spelled out in the Methods.

This is now spelled out where it first appears.

7. Figure 6 Legend, Line 551: Remove "and" in "...irrespective of and disease elements..."

This has been removed.

8. Figure S2 Legend, Line 565: "The frequency [add "of"] known APEC-associated..."

This has been added.

Reviewer #2 (Remarks to the Author):

1. The study represents a valuable comparative genomic analysis aiming at the identification of pathogenicity markers of APEC. A major advantage of the study is the inclusion of disease-associated isolates from different body (infection) sites and of asymptomatic faecal isolates in the analysis.

We thank the reviewer for their summary and positive comments. We respond to specific comments below.

2. The design of the study is convincing with regard to the analysis of the phylogenetic diversity of avian *E. coli* and the prediction of genomic differences between disease-associated and asymptomatic faecal colonizers. However, against the background of the title of the manuscript, the data presented are not convincing in terms of statements on "evolution and emergence" of pathogenicity.

Consistent with the reviewers comment we have amended the title to: 'Genome evolution and the emergence of pathogenicity in avian Escherichia coli'. This emphasizes the focus on the nature and dynamics of micro-evolutionary events within the genome.

3. The authors confirmed once more that APEC can derive from several major phylogenetic lineages. The description of 143 disease-associated genes that appear not to be lineage-specific points towards promising candidates. In order to confirm this assumption, which is so far only based on comparative genomic prediction, and to support the suggestion that these "markers" could indeed promote risk prediction, experimental evidence for their role in pathogenicity is required. It should be experimentally tested and explained how, e.g. certain allelic variants of house keeping genes or O antigen biosynthesis genes contribute to increased pathogenicity.

*We welcome this comment and fully agree that statistical genetic validation is not the same as the traditional microbiological validation needed for full functional genomic understanding. Transformative population-wide genomic screening/analysis approaches, such as GWAS and machine learning, deliver a deluge of potentially useful information describing the genetic basis of complex traits. If correctly integrated with lab microbiology, this can underpin rigorous confirmatory tests, such as Molecular Koch's postulates¹. However, in extremely diverse organisms like pathogenic *E. coli*, describing a genotype-phenotype map is really a community enterprise rather than something that can be delivered by individual labs. While we are currently working to knock out several of the pathogenicity-associated genes in APEC, we have preferred to publish our findings in a timely manner so that all the genome data and analysis can be made available to inform future studies by ourselves and others. To reiterate, however, we do appreciate the need for more detailed functional genomic understanding and have added text to this effect in the revised submission (Lines 310 - 317, 356 - 360 and 440 - 442).*

4. It is understandable that in the search for pathogenicity relevant markers of APEC only avian isolates were analyzed and it is greatly appreciated that the authors included a large and diverse group of strains into their study. However, in the context of analyzing potential disease-associated markers, it would be important to know that the predicted markers do not also occur in other *E. coli* pathotypes. This aspect is missing in the study so far. It will be important to show that the predictors that result from the GWAS and machine learning approaches are actually discriminatory for APEC. Can such alleles also be found, for example, in human strains or in extraintestinal pathogenic *E. coli* isolates from cats and dogs?

*We agree that it would be interesting to contextualise our findings with other *E. coli* pathotypes, especially outside of the avian niche. While this goes some way beyond the scope of this study, we have conducted additional analyses to quantify the prevalence of APEC-associated SNPs in the *E. coli* Reference Collection (ECOR), as well as in 3 other *E. coli* collections: (i) 175 human ExPEC strains²; (ii) 14 disease-associated strains from dogs³; (iii) 521 strains from healthy cattle⁴. In many cases the same genetic variants occur in isolates from other sources implying that host species does not constitute a complete gene-pool barrier. A Kruskal-Wallis test identified significant differences in the*

presence of genetic markers in human ExPEC compared to the ECOR collection ($p < 0.0001$) and healthy bovine isolates ($p < 0.0001$), indicating a difference in the prevalence of APEC-associated genetic variants (Figure S5B). However, some APEC-associated variants were also common in isolates from other sources, especially human ExPEC. This may suggest common adaptations to establishing extraintestinal infection in both avian and human hosts. The percentage of each APEC-associated variant used in the RF model in each of these additional *E. coli* groups is shown in Figure S5C. We present these new results in Figure S5/Table S6 and discuss them at lines 242 - 255, 581 - 588 and 689 - 699.

5. The authors did not find many "typical" *E. coli* virulence genes, such as adhesins, T3SS, T5SS, T6SS secretion systems, etc., but many house keeping and metabolic genes. As the authors themselves write in the introduction, *E. coli* infections in poultry are very diverse and result from different infection routes. Accordingly, aerocolitis develops and manifests differently from omphalitis or salpingitis. The route of infection could indicate a dependence on certain virulence factors that are required for this. The selectivity of the analyses presented in the manuscript could suffer from this in the same way as from the fact described by the authors that the pathogenicity potential of asymptomatic colonizers of the intestinal tract cannot be predicted exactly. To increase the selectivity, GWAS analyses should not only be performed with different phylogroups, but also with isolate groups distinguished on the basis of the type of infection, as the mechanisms and routes of infection or colonization seem to play an important role.

Multiple other configurations of GWAS would be very interesting to perform. However, a trade-off exists between the resolution power of GWAS and the sample size of groups examined. In this study, we chose to perform 4 distinct GWAS in the 4 most common E. coli phylogroups. That way we were able to obtain statistically significant results for each GWAS and interrogate their biological relevance as generic global markers of pathogenicity (for all APEC) by considering hits that were consistently present in all the GWAS experiments. The possibility of exploring genetic markers that are associated with specific infection types would be an exciting study and it will be more feasible in the future when greater numbers of strains from various infection locations have been isolated and sequenced. Nonetheless, to address the reviewer's enquiry, we have conducted additional analysis to plot the distribution of associated elements in isolates from different sample sources (Figure S5A/Table S7). A Kruskal-Wallis test for significance revealed significant differences between asymptomatic carriage and bone marrow samples ($p < 0.0001$), between asymptomatic carriage and liver isolates ($p = 0.0003$) and between bone marrow and heart isolates ($p = 0.009$). This has been discussed in the text of the resubmission (183 - 184, 212 - 213, 257 - 262, 326 - 334, 339 - 344, 398 - 404 and 689 - 699).

6. Why has *E. coli* O157 been used as a reference for genome-based phylogeny of APEC?

The choice of reference genome can introduce bias in variant calling via read mapping for species with high intragenomic diversity⁵, although these are unlikely to have an impact on the resulting phylogeny⁶. We chose a well-characterised closed reference genome, APEC O157, from Phylogroup E, which was absent from the collection under investigation in order to minimise and standardise any bias caused by reference strain selection. As all isolates were mapped to a single reference, this also allowed for a tree to be constructed from all isolates and comparison of genomic/alignment regions used for tree building during the recombination analysis. O157 used as the reference for phylogenetic reconstruction and other references, including APEC_01, were included in the pan genome analysis in order to provide points of reference for comparison of genes to well-characterised reference genomes. This has now been clarified in the text (Lines 497 - 506).

7. Many plasmid-associated genes can also be located on the bacterial chromosome, e.g. in genomic islands (typical examples are siderophore determinants, e.g. coding for aerobactin or salmochelin, autotransporter genes, T3SS gene clusters). How can the authors distinguish between chromosomal and plasmid genes? BLAST searches alone are not really helpful. For a proper allocation of genes to either the chromosomal or to plasmid(s), their sequence context would have to be considered as well. Accordingly, the statements regarding plasmid genes should be cautiously interpreted. Wouldn't it be also helpful to assess the prevalence and diversity of plasmid types in APEC? The authors should include pMLST and incompatibility group-typing of the plasmid-derived contigs in the genome sequences.

We appreciate these comments and recognize the challenge of discriminating between chromosomal and plasmid genes. Indeed, this has been a specific research area within the Sheppardlab for some time^{7,8}. Based upon our experience, robust inference requires that analysis is performed separately on miniprep extractions that are purified multiple times, to maximise the likelihood of sequencing only

plasmid-borne DNA. In ongoing work, we are seeking to characterize APEC plasmids using long read sequencing (MinION) to quantify the degree to which: (i) known plasmids represent discrete elements, (ii) genes are found in multiple plasmid backgrounds, and (iii) particular genes are obligate plasmid-borne elements or are chromosomally integrated. However, this is a considerable research endeavour when applied population-wide in diverse species such as *E. coli*.

In this study, we do not focus on discrete plasmids but on the mosaic of possible plasmid and chromosomal genes that have been reported in APEC. Our findings (Figure 2 and S2) are consistent with extensive recombination between plasmids in avian *E. coli* as shown in previous literature⁹ and we agree with the reviewer that the context of genes that have been reported in plasmids is interesting. However, the main aim of our study is to deliver an agnostic statistical approach that centres on the identification of population-wide genetic elements associated with ecological traits, before returning to more detailed investigation of the type of variation and the putative gene function. We hope that our findings will be integrated with wider community endeavour to understand the genetic basis of pathogenicity in *E. coli*. Consistent with this, and the reviewers comments, we have amended the revised submission to:

(i) clarify our gene-centred approach and definition of plasmidic genes - as those that have been found to be harboured in a plasmid from an APEC strain in the literature, whether the plasmid is integrated in the chromosome or not (lines 531 - 539).

(ii) exercise caution when interpreting genes as plasmid-born (now described as putative throughout) and mention that additional analysis are necessary to prove genes are plasmidic (lines 138 - 139).

(iii) include additional analysis of pMLST and incompatibility group-typing of the plasmid-derived contigs, using PlasmidFinder 2.1 database¹⁰ (Table S5). We note that typing plasmids that are possibly subject to such extensive recombination (mosaicism) should be treated with caution and have described potential plasmid genes identified in our gene-by-gene analysis as 'putative' throughout the revised manuscript (lines 531 - 539).

8. How relevant and meaningful are the statements in lines 287-299 with regard to the generated data and their interpretation regarding the ability of APEC to cause infection in contrast to asymptomatic colonizers? The *hokA* and *hokC* genes coding for components of toxin-antitoxin systems are certainly not APEC virulence determinants in the strict sense, but indicate the presence of plasmids or derived mobile genetic elements (on which possibly much more relevant genes could be located...).

Why do the authors find only the chaperone-encoding genes *papD* and *fimD* among the virulence-associated genes and not the entire P or type 1 fimbrial determinants? Is the assembly of P- and type 1 fimbriae affected by these selected chaperone alleles?

Which APEC toxins should be secreted by the general secretion pathway? For typical toxins of extraintestinal pathogenic *E. coli* (e.g. haemolysin and different cyclomodulins) the general secretion pathway does not play a mechanistic role. Evidence of a specific advantage from the presence of these gene variants in APEC is missing. The mutation rate and possible selection conditions have not been tested, which could provide alternative indications of possible allele variations independent of diseases association.

We welcome these comments and the opportunity to improve explanations of the GWAS approach and results. Our work is not intended as a definitive road-map to describing APEC pathogenicity, rather it is an agnostic analysis of the largest APEC dataset to date, with the goal to describe generalised pathogen-associated microevolutionary events and present a realistic statistical genetic platform upon which understanding of genome evolution and ongoing studies such as risk analysis can be built. We accept that full functional genomic understanding of pathogenesis will require further microbiological validation (lines 310 - 317). However, while large numbers of known pathogenicity genes were flagged in our population-wide study, it is inevitable that the findings will not exactly match with data from existing studies for two important reasons.

First, a total of 143 genes were found to contain pathogenicity-associated elements. This may be unsurprising for such a complex multifactorial phenotype but there is a requirement to rank these based upon the significance score as there are so many. This will inevitably flag more wide-spread pathogenicity elements (consistent with our aims) but variation that is associated with a specific infection type, such as aerosacculitis, omphalitis or salpingitis, may have lower significance. For example, while the *hokA* and *hokC* genes coding for components of toxin-antitoxin systems are not APEC virulence determinants in the strict sense, they may indicate the importance of wide-spread mobile genetic elements that are linked to virulence genes that vary by infection type (clarified and explained at lines 341 - 347).

Second, many of the most significant hits were in core genes. Specifically, of the 79 species-wide predictors used to train the RF model, 66 were SNPs in core genes and 3 represented accessory gene presence/absence (Table S3). This has implications for the findings. For example, while the chaperone-encoding genes *papD* and *fimD* are among the pathogenicity-associated genes other *P* and type 1 fimbrial determinants are not, despite their likely role in virulence. The reason for this is that the variation within the genes encoding fimbrial determinants does not segregate as strongly as the chaperone genes based on the binary pathogenicity phenotype in this study. This suggests that multiple homologous sequence variations within these genes can underlie pathogenicity when particular SNPs are present in the chaperone genes. For instance, it is known that the *fimD* gene required for cell surface localization of type 1 fimbriae¹¹ and *Pap* proteins have been found to be located in the outer membrane in both pathogens and commensals¹². Furthermore, we also found cases of gene duplication + fission/fusion in *fimD* so it may be that expression levels in chaperone genes can be important in virulence but further functional genomic characterization is necessary to confirm this (clarified at lines 347 - 356). Aside from polymorphism in particular fimbrial regions, it is commonly accepted that factors promoting host colonization can sometimes be considered virulence factors as well, as attachment and colonization are often seen as the first step to bacterial pathogenicity, including of ExPEC¹³⁻¹⁵. However, these factors, or truncated fractions of their genes¹⁶ can also be shared between pathogenic and non-pathogenic/commensal strains from related taxa, sometimes conducive to commensal persistence in the gut. This does not preclude adhesion factors from having a specific role in the virulence of extra-intestinal pathogenic strains, but would however impact their increased statistical enrichment in pathogenic strains, and the subsequent ability by statistical methods to identify them as being specific to pathogens. The re-submission has been substantially revised to contain improved descriptions, with examples, of how to interpret the statistical associations in the context of the wealth of functional genomics research on APEC (Lines 356 - 360).

9. The suggestion that the presence of accessory disease-associated genes on mobile elements represents an evolutionary advantage, because it allows rapid accumulation and reassortment to suit changing environmental conditions is interesting. Do the authors know gene transfer rates in the intestinal tract or in extraintestinal Habitats? What is known about the time frame of extraintestinal infection in poultry to enable such processes? Shouldn't the authors assume chronic extraintestinal infections to allow for virulence determinants in APEC to accumulate and reassort to this extent?

We appreciate this comment. With evidence for the divided genome and re-assortment through elevated HGT among putative pathogenicity genes (Figure 4) as the core message, considering the timescale for transfer is very interesting. However, different recombination rates among genes (pathogenicity vs core) and a varying role for selection would confound our usual model-based approaches to determining recombination rate¹⁷. Considering the location and time-scale of gene transfer is extremely interesting. However, our findings suggest a recombining population, with large effective population size, at the interface between the commensal environment and systemic tissues. Extraintestinal spread, potentially mediated by multiple factors, leads to a genetic bottleneck and favouring certain strains which proliferate. Consistent with the reviewers comment we now consider the time-scale for HGT in the context of chronic extraintestinal infections that may allow virulence determinants to accumulate and re-assort among strains (Lines 386 - 391).

Reviewer #3 (Remarks to the Author):

1. This study by Mageiros and colleagues examines the genetic elements that distinguish avian pathogenic *E. coli* (APEC) from commensal *E. coli* located in the avian gut. It also discusses the likely evolutionary processes that dominate the shift between the invasive and commensal states, arguing that horizontal gene transfer is the dominant mechanism. Finally, the study presents a random forest model for predicting pathogenicity based on a limited set of genetic markers. The study is methodologically solid and the claims are well founded. It is not groundbreaking in terms of methodology. In fact, it is almost a step-by-step recreation of Méric and Mageiros et al, 2018 (<https://www.nature.com/articles/s41467-018-07368-7>), only with a different organism. APEC is nearly an ideal choice of organism for such a study though, as it is known to emerge after acquiring pathogenicity elements, is a high-impact disease on welfare, production and economy, and might be preventable if its emergence was better understood. The central ideas of the paper are presented in a logical order and makes for an entertaining read. The conclusion that HGT is a more dominant mechanism for the emergence of APEC than plasmid transfer is original, and might be somewhat controversial, because the presence of various plasmids has been seen as the defining characteristic of APEC and a solid way to distinguish one from the other.

We thank the reviewer for their positive remarks about the potential importance and novel (if potentially controversial) findings. We did not set out to be challenging, rather we take an agnostic approach that relies on a large sample collection and a statistical approach to identifying associations. We hope that the findings, that a divided genome evolutionary model explains emergence better than strict plasmid transfer (although they are important), will be a platform for future attempts to understand and control APEC.

2. For me, the major weakness of the paper is that there is too little focus on putting its findings into context with existing literature. For another example, the paper presents a list of genes that were most highly associated with pathogenicity. (Highlights in the results section and full list in supplementary). This list has little overlap with "established" APEC virulence factors. For example, the paper cites Delicato et al (2003), which posits that iutA, iss, cvaC, tsh, papC, papG and felA are virulence genes. None of these are found in the results here and the results are not discussed. Similarly, Janben et al (2001) is cited, but the virulence genes found in that paper is not put into context of the current manuscript (iucD, tsh, fimC, fyuA and irp2). A very recent resource is the textbook chapter on colibacillosis by Nolan et al 2020 (<https://doi.org/10.1002/9781119371199.ch18>), which lists a large number of virulence genes ordered by different functions. Again, there is not much overlap between these virulence factors and the ones presented from the GWAS model of the current paper.

We appreciate this comment. While the main novelty of the paper is describing the divided genome model of APEC emergence, we welcome the opportunity to improve the contextualization of associations and explain the incomplete overlap with established virulence factors. At lines 336 - 360 in the re-submission we have:

(i) Substantially revised and augmented the contextualization of APEC-associated genes. This Includes studies with no overlap with our associated genes¹⁸ as well as those where there is a link. For example, we identified fimD as an important gene which, while not specifically highlighted, is associated with type 1 fimbriae that are extensively discussed in published studies relating to E. coli virulence^{19,20}. We particularly appreciate the reviewer's suggestion to include authoritative Nolan et al. (2020) paper, published after the submission of our manuscript. In the revised submission, we cross-reference this paper with our results and identify two genes that are APEC associated in both studies. Specifically, eae - an attaching and effacing gene that encodes intimin, and ompT - that encodes a protease able to cleave colicin. Additionally, we detected genes that belong to the same operons as other genes reported by Nolan et al., including fimD, papD and tral.

(ii) Provided a detailed description of the reasons why findings from population-wide GWAS approach will not exactly match with data from existing studies. Rather our approach highlights population-wide genetic variation that is associated with multiple pathologies and segregates by the binary phenotype (ie commensal vs pathogenic strains). This is discussed in detail in response 8 to reviewer 2.

3. The manuscript presents a very interesting idea, that some strains can be "predisposed to pathogenicity". This is first mentioned in the introduction and then again at the end of the results and in the discussion. But if this is a goal of the paper, it is never fully achieved. It would be very useful if a set of sufficient (or necessary) conditions for pathogenicity could be found. (True, this would be a daunting task.)

We agree that this is a daunting task, but a worthy one none the less. In this context, the predisposition to pathogenicity can be considered as belonging to those strains where the genome harbours elements that confer virulence and related functions. A typical view of such strains is that they acquire specific genomic elements and then spread, as in some pandemic ExPEC clones. APEC has proved resistant to such simple evolutionary models and it was/is our aim to take a different approach. Rather than developing a definitive road-map to pathogenicity, which is essentially a microbiology community enterprise, we take an agnostic analysis approach to a very large APEC genome dataset. The goal is to better understand APEC genome evolution and describe elements associated with multiple lineages and pathologies. Thus, providing a view of natural population genomic variation and realistic platform upon which full functional genomics validation can build. This has been clarified in the revised submission (lines 310 - 317 and 356 - 360).

4. The manuscript mentions as complicating factors lineage-specific and tissue-specific virulence factors. I would have liked to see this idea explored further. For example, it is clear that the authors have information on the extraintestinal location of APEC isolates, but this information does not appear to have been included in the GWAS analysis?

This recommendation is consistent with comment 5 of reviewer 2. We have appreciated the opportunity to add additional analysis. As the statistical power of the GWAS increases with the number of isolate

genomes, a binary division (pathogenicity vs commensal) was preferred over more specific groupings (different extraintestinal locations). This returned generic global markers of pathogenicity but potentially misses pathology specific ones. Nonetheless, to address the reviewer's enquiry, we have conducted additional analysis to plot the distribution of associated elements in isolates from different sample sources (Figure S5/Table S7). A Kruskal-Wallis test for significance revealed significant differences between asymptomatic carriage and bone marrow samples ($p < 0.0001$), between asymptomatic carriage and liver isolates ($p = 0.0003$) and between bone marrow and heart isolates ($p = 0.009$), suggesting that multiple infection routes could co-exist for APEC. This has been briefly discussed in the text of the resubmission (183 - 184, 212 - 213, 257 - 262, 326 - 334, 339 - 344, 398 - 404 and 689 - 699).

5. As for lineage-specific virulence factors, that might indeed make the whole thing more complex. However, the authors' own RF model drops only about 3% from training on the entire set to training on A, B1 and B2, and testing on ST-117. As they correctly point out, that means that a limited set of global pathogenicity markers must be the driving forces. In the RF model, a majority of the predictors are core genome SNPs in hypothetical proteins. This makes it seem likely that rather than predicting APEC/commensalism from virulence factors, the model is probably predicting some kind of phylogenetic signal, or mutations that are hitchhiking with real virulence factors in HGT.

The RF predictors are genetic variants that overlap in 4 distinct GWAS's on groups of genomes from APEC vs. commensal *E. coli*. The GWAS approach rigorously accounts for phylogenetic structure and the RF predictors perform well in the whole dataset but also when we remove one phylogroup from the training procedure and use it as our prediction set. While we welcome the comment, we think that among the best evidence that phylogenetic position is not the driving force (indicated by the GWAS and RF) is the evidence of convergent evolution (homoplasy) among the associated elements (lines 314 - 317 and 376 - 380). However, the reviewer mentions an interesting point about genetic hitchhiking with 'real' virulence factors. Genomes are co-adapted landscapes in which genes covary not only because of the clonal frame but also because they are evolving within the same functional interaction network. Untangling this web is generally very difficult without functional genomic characterization in the lab (currently being carried out for several genes). However, covariation has a limited confounding impact on our aim to flag pathogenicity risk predictors, whether they include all causal functional network components or not. Consistent with the reviewer's line of questioning, this has been clarified in the revised submission (lines 336 - 356 and 429 - 433).

Below I have comments to specific parts of the manuscript:

6. - Availability of data: Looking at bioproject accession PRJNA592536 there seem to be missing some data. That accession only contains data from 259 isolates. The paper refers to 568 isolates, and from Supplementary table S1 it's clear that 414 are from "This study".

The bioproject for this study has been updated to contain all 414 isolates sequenced in this study and the accession numbers for other isolates are included in Table S1 to a total of 568.

7. - Please double check that all the numbers in the manuscript are correct and correspond to those of the tables. For example, I have found the following inconsistencies:

We are grateful that the reviewer pointed this out. A thorough manuscript check has confirmed that text and table numbers now correspond.

(1) - In line 158, 220 isolates are said to be ST-117. However, in the supplementary table, only 211 are listed.

This has been corrected and column E in Table S1 now has 220 entries. This was because there are 211 isolates that are characterised as ST-117 but 220 isolates that belong to the ST-117 clonal complex.

(2) - In the sampling section of the materials and methods, isolate numbers by host: $475 + 33 + 12 + 5 = 525$. But the total is supposed to be 568?

The remaining isolates were collected by an industrial partner. We have now obtained this isolate source information and added it to Table S1 and updated the text (lines 449 - 452). This includes samples from chickens, farmers' boots and one from an unspecified source. This has also been explained in the text.

(3) - In the same section, 243 were from asymptomatic, 307 were APEC. What about the final 18? Also, Suppl table 1 only lists 242 as asymptomatic.

In some cases, it was difficult to robustly confirm isolate disease association. With the additional source information added, we have corrected the isolate numbers associated with asymptomatic carriage, infection and unknown source in the text and tables (Table S1, lines 452 - 459).

8. - Results, line 178-182 - The number of discovered elements with $p < .05$ is very high - Over 43,000 for B2 alone! Over all the major phylogroups, the significant GWAS results links 5200 genes to invasiveness. This seems like an extraordinarily high number, seeing as the total pan-genome is around 15,000 genes for this collection. (Evident from suppl fig 1).

Data are presented for the relaxed p-value threshold of $p < 0.05$ for all GWAS's and, as stated, there are a large number of hits. These data are included, in part, to illustrate the magnitude of variation and are possibly inflated by lineage effects, some false positives etc. Importantly, however, downstream analysis uses $p < 0.01$, as does Figure 3B. At this threshold we detect 896 pathogenicity-associated genes overall (753 lineage-specific and 143 pathogenicity-associated in all 4 GWAS runs). We are grateful for the opportunity to make this clearer in the revised submission (Lines 177 - 181).

9. Now, looking at figure 3B, it appears as if the number of genes included in the analysis is roughly 10,500.

As mentioned above, there are 896 significantly associated ($p < 0.01$) pathogenicity genes plotted (dots) in Figure 3B, which include both lineage-specific and APEC-wide hits. The inner circle corresponds to $p < 0.01$ with concentric rings emanating from this threshold corresponding to incremental reductions to a p-value of 0.000001 (9th ring). The numbers on the outer ring denote the length of the pangenome in Mbp. This has been clarified in the revised the figure legend and text (lines 180 - 181).

10. Does this mean that half of all genes had some variant linked with pathogenicity? That seems rather extreme. How many of these are simply phylogenetic variants that predict a single invasive cluster?

A total of 896 genes have been identified to harbour genetic elements significantly associated with pathogenicity ($p < 0.01$). Of these, 143 are consistently found in all of the different GWAS iterations performed. This has been clarified at lines 184 - 185.

11. In the following sentence, 753 of these have $p < .01$, and exactly the same number are found to be lineage-specific? Is this a coincidence or is there a numbering error here?

Many thanks for spotting this error. It has been corrected (lines 181 - 184).

12. - 143 genes are identified as significant in the GWAS model across all phylogroups. From these candidate genes, the list is trimmed to 79 predictor elements by selecting only those with complete(?) segregation between commensal and pathogenic. It's unclear to me why such a strict criteria was imposed, because it means that the predictor list does not contain any elements from many of the genes that were predicted in the GWAS to be most associated with disease. For example, the top GWAS hit, yeeO, has no associated elements in the predictor list.

Focussing on associated elements from multiple infection types and phylogroups maximises the likelihood of identifying ubiquitous markers that discriminate APEC from commensal E. coli, and limits possible confounding by lineage- or pathology-specific determinants. The significant hits are interesting and warrant further investigation. However, the more stringent parts of our pipe-line are aimed at identifying global predictors. This has been clarified at lines 192 and 212 - 213.

13. - Lines 198-200 - 0.156 unique alleles per locus for the 143 pathogenicity-associated genes. - I would have liked to have an explanation in the methods section of how this metric is calculated because I don't understand it. How is there less than one unique allele per locus? How can any genes have zero number of unique alleles per locus? (Fig 4) –

We defined the number of alleles per locus as the number of unique alleles per gene divided with the total number of isolates. An explanation of this point has been added in the Methods section at lines 555 - 557.

14. - Supplementary table S1 - It would be nice if the individual run accessions were listed for each isolate.

This has now been updated in Table S1.

15. - Supplementary table S1 - Some metadata missing. I'm sure the authors know what year and country their own isolates hail from, and this should be entered in the table.

All available metadata has now been added to Table S1.

16. - Supplementary table S1 - Some hard to interpret metadata. What is "HTP"? What is "boot drag"?
Table S1 has now been updated and clarified. "HTP" was a former code for isolation, corresponding to an isolate swabbed from a at the end of their use. We clarified this and rephrased it as "Farm environment".

17. - Phylogenetic analysis - Pseudoreads were generated from assembled genomes. However, there are no details on how the genomes were assembled initially nor how the pseudoreads were generated. *Briefly, pseudo-reads were created as a part of the SNIPPY pipeline used for variant calling, in which contigs are first split into 250bp single-end read pairs at a simulated ~20x coverage of the reference genome. The methods section has been updated with an explanation (lines 502 - 506).*

18. - Fig 3, panel B - The outside ring. What is the unit? Is it gene number or does it correspond to genome coordinates?

The numbers in the outer ring denote the gene position in the pangenome (Mbp). The legend of Figure 3B has been updated to clarify this.

References

- 1 Falkow, S. Molecular Koch's postulates applied to microbial pathogenicity. *Rev Infect Dis* **10 Suppl 2**, S274-276, doi:10.1093/cid/10.supplement_2.s274 (1988).
- 2 McNally, A. *et al.* Combined Analysis of Variation in Core, Accessory and Regulatory Genome Regions Provides a Super-Resolution View into the Evolution of Bacterial Populations. *PLoS Genet* **12**, e1006280, doi:10.1371/journal.pgen.1006280 (2016).
- 3 Valat, C. *et al.* Pathogenic *Escherichia coli* in Dogs Reveals the Predominance of ST372 and the Human-Associated ST73 Extra-Intestinal Lineages. *Front Microbiol* **11**, 580, doi:10.3389/fmicb.2020.00580 (2020).
- 4 Arimizu, Y. *et al.* Large-scale genome analysis of bovine commensal *Escherichia coli* reveals that bovine-adapted *E. coli* lineages are serving as evolutionary sources of the emergence of human intestinal pathogenic strains. *Genome Res* **29**, 1495-1505, doi:10.1101/gr.249268.119 (2019).
- 5 Bush, S. J. *et al.* Genomic diversity affects the accuracy of bacterial single-nucleotide polymorphism-calling pipelines. *GigaScience* **9**, doi:10.1093/gigascience/giaa007 (2020).
- 6 Lee, R. S. & Behr, M. A. Does Choice Matter? Reference-Based Alignment for Molecular Epidemiology of Tuberculosis. *J Clin Microbiol* **54**, 1891-1895, doi:10.1128/jcm.00364-16 (2016).
- 7 Meric, G. *et al.* Lineage-specific plasmid acquisition and the evolution of specialized pathogens in *Bacillus thuringiensis* and the *Bacillus cereus* group. *Mol Ecol* **27**, 1524-1540, doi:10.1111/mec.14546 (2018).
- 8 Nguyen, L. T. T. *et al.* The emergence of plasmid-borne cfr-mediated linezolid resistant-staphylococci in Vietnam. *J Glob Antimicrob Resist* **22**, 462-465, doi:10.1016/j.jgar.2020.04.008 (2020).
- 9 Boyd, E. F., Hill, C. W., Rich, S. M. & Hartl, D. L. Mosaic structure of plasmids from natural populations of *Escherichia coli*. *Genetics* **143**, 1091-1100 (1996).
- 10 Carattoli, A. *et al.* In silico detection and typing of plasmids using PlasmidFinder and plasmid multilocus sequence typing. *Antimicrob Agents Chemother* **58**, 3895-3903, doi:10.1128/AAC.02412-14 (2014).
- 11 Klemm, P. & Christiansen, G. The fimD gene required for cell surface localization of *Escherichia coli* type 1 fimbriae. *Mol Gen Genet* **220**, 334-338, doi:10.1007/BF00260505 (1990).
- 12 Korea, C. G., Ghigo, J. M. & Beloin, C. The sweet connection: Solving the riddle of multiple sugar-binding fimbrial adhesins in *Escherichia coli*: Multiple *E. coli* fimbriae form a versatile arsenal of sugar-binding lectins potentially involved in surface-colonisation and tissue tropism. *Bioessays* **33**, 300-311, doi:10.1002/bies.201000121 (2011).
- 13 Law, D. Adhesion and its role in the virulence of enteropathogenic *Escherichia coli*. *Clin Microbiol Rev* **7**, 152-173, doi:10.1128/cmr.7.2.152 (1994).
- 14 Hagberg, L. *et al.* Adhesion, hemagglutination, and virulence of *Escherichia coli* causing urinary tract infections. *Infect Immun* **31**, 564-570, doi:10.1128/IAI.31.2.564-570.1981 (1981).
- 15 Mainil, J. *Escherichia coli* virulence factors. *Vet Immunol Immunopathol* **152**, 2-12, doi:10.1016/j.vetimm.2012.09.032 (2013).
- 16 Zdziarski, J., Svanborg, C., Wullt, B., Hacker, J. & Dobrindt, U. Molecular basis of commensalism in the urinary tract: low virulence or virulence attenuation? *Infect Immun* **76**, 695-703, doi:10.1128/IAI.01215-07 (2008).
- 17 Mourkas, E. *et al.* Agricultural intensification and the evolution of host specialism in the enteric pathogen *Campylobacter jejuni*. *Proc Natl Acad Sci U S A* **117**, 11018-11028, doi:10.1073/pnas.1917168117 (2020).
- 18 Delicato, E. R., de Brito, B. G., Gaziri, L. C. & Vidotto, M. C. Virulence-associated genes in *Escherichia coli* isolates from poultry with colibacillosis. *Vet Microbiol* **94**, 97-103, doi:10.1016/s0378-1135(03)00076-2 (2003).

- 19 Janben, T. *et al.* Virulence-associated genes in avian pathogenic *Escherichia coli* (APEC) isolated from internal organs of poultry having died from colibacillosis. *Int J Med Microbiol* **291**, 371-378, doi:10.1078/1438-4221-00143 (2001).
- 20 Klemm, P. & Hedegaard, L. Fimbriae of *Escherichia coli* as carriers of heterologous antigenic sequences. *Res Microbiol* **141**, 1013-1017, doi:10.1016/0923-2508(90)90143-e (1990).

REVIEWERS' COMMENTS

Reviewer #1 (Remarks to the Author):

My previous comments have been addressed sufficiently in the revised version of this manuscript. I am happy with the changes made.

Reviewer #3 (Remarks to the Author):

Review of "Genome evolution and the emergence of pathogenicity in avian *Escherichia coli*" by Mageiros and colleagues, 1st resubmission.

NOTE: For a more general description of this work and statements about its potential impact, please see my review of the original submission.

I would like to thank the authors for giving thorough answers to all my questions and taking all my raised concerns into account when amending their manuscript.

In my first review, my major concern was that the manuscript did not put findings into context with existing knowledge on APEC. The current revision includes a solid discussion on differences between these findings and determinants known from previous literature. I also mentioned that I would have liked to see tissue-specific analyses explored more thoroughly. The current revision includes this and as a bonus has a very nice figure in the supplementary material. My comments on specific errors in the manuscript including tables/figures have all been addressed.

I have a few more comments:

- Abstract: "Finally, a random forest model prediction of disease status (carriage vs disease) identifies pathogenicity elements in the emergent ST-117 poultry-associated lineage with 73% accuracy" - This seems wrong. Doesn't the 73% accuracy refer to the prediction of carriage/disease?

- Lines 221-224: "SNPs within the *gnd* gene [...] accounted for 4 of the 10 most important predictors which achieved a classification accuracy of 77.5% on their own [...]" - I think I must be missing some context here. Did 4 SNPs in the *gnd* gene alone give better out-of-sample classification accuracy than all 79 predictors (76.9%)? Or (alternative interpretation), do the top 10 predictors give 77.5% accuracy? In any case, this is curious. Why then, are the remaining predictors even included in the RF model? On lines 189-192 it is stated that all 79 predictors segregate disease/carriage with $p < 0.05$. I'm struggling to think of a scenario where including such markers would lower accuracy. Not sure whether the authors have made a mistake here or if this is just a confusing result of the model. If it is the latter, perhaps this confusing result should be mentioned?

- Materials and methods: "One isolate was from the poultry farm environment" - This strain (8649?) has "Isolation source" set to "Broiler chicken" in Table S1. I would recommend that to be changed.

- Could the authors clarify which parts of the study the environmental strains are used for? E.g. pangenome characterization, GWAS, phylogenetics + figures, RF classifier etc. If they are in the figures they should have a separate colour.

RESPONSE TO REVIEWERS

Reviewer #1 (Remarks to the Author):

My previous comments have been addressed sufficiently in the revised version of this manuscript. I am happy with the changes made.

We thank the reviewer for their comments and we are happy that we were able to satisfy all of them.

Reviewer #3 (Remarks to the Author):

I would like to thank the authors for giving thorough answers to all my questions and taking all my raised concerns into account when amending their manuscript.

In my first review, my major concern was that the manuscript did not put findings into context with existing knowledge on APEC. The current revision includes a solid discussion on differences between these findings and determinants known from previous literature. I also mentioned that I would have liked to see tissue-specific analyses explored more thoroughly. The current revision includes this and as a bonus has a very nice figure in the supplementary material. My comments on specific errors in the manuscript including tables/figures have all been addressed.

We are grateful for the reviewer's comments. Having addressed these we consider the manuscript to be improved considerably.

I have a few more comments:

- Abstract: "Finally, a random forest model prediction of disease status (carriage vs disease) identifies pathogenicity elements in the emergent ST-117 poultry-associated lineage with 73% accuracy" - This seems wrong. Doesn't the 73% accuracy refer to the prediction of carriage/disease?
We welcome this comment and have updated the Abstract accordingly (line 43).

- Lines 221-224: "SNPs within the *gnd* gene [...] accounted for 4 of the 10 most important predictors which achieved a classification accuracy of 77.5% on their own [...]" - I think I must be missing some context here. Did 4 SNPs in the *gnd* gene alone give better out-of-sample classification accuracy than all 79 predictors (76.9%)? Or (alternative interpretation), do the top 10 predictors give 77.5% accuracy? In any case, this is curious. Why then, are the remaining predictors even included in the RF model? On lines 189-192 it is stated that all 79 predictors segregate disease/carriage with $p < 0.05$. I'm struggling to think of a scenario where including such markers would lower accuracy. Not sure whether the authors have made a mistake here or if this is just a confusing result of the model. It is the latter, perhaps this confusing result should be mentioned?

Thank you very much for spotting this inconsistency. Indeed, this number was a typing error and the correct value is 73.5% for all the top 10 predictors. The text has been corrected and clarified in line 244.

- Materials and methods: "One isolate was from the poultry farm environment" - This strain (8649?) has "Isolation source" set to "Broiler chicken" in Table S1. I would recommend that to be changed.
This is the strain with id number 8649. This strain has now the value "Farm Environment" in the filed isolation source in Supplementary Data 1.

- Could the authors clarify which parts of the study the environmental strains are used for? E.g. pangenome characterization, GWAS, phylogenetics + figures, RF classifier etc. If they are in the figures they should have a separate colour.

The environmental strains indicated in Supplementary Data 1 do not have a phenotype characterisation assigned to them in the 'Population' column as the origin was not a discrete source. Therefore, they were only used in the in the phylogenetic and pangenome analysis of this study and they do not appear in any figures of the downstream analysis. This has been clarified in lines 460 – 462 of the resubmission.